# Omega-3 fatty acid epoxides produced by PAF-AH2 in mast cells regulate pulmonary vascular remodeling

Hidenori Moriyama[1], Jin Endo [1✉], Masaharu Kataoka[1], Yuta Shimanaka [2], Nozomu Kono [2], Yuki Sugiura[3], Shinichi Goto[1], Hiroki Kitakata[1], Takahiro Hiraide[1], Naohiro Yoshida[1], Sarasa Isobe [1], Tsunehisa Yamamoto[1], Kohsuke Shirakawa[1], Atsushi Anzai [1], Yoshinori Katsumata[1], Makoto Suematsu [3], Kenjiro Kosaki[4], Keiichi Fukuda [1], Hiroyuki Arai[2] & Motoaki Sano[1]

Pulmonary hypertension is a fatal rare disease that causes right heart failure by elevated pulmonary arterial resistance. There is an unmet medical need for the development of therapeutics focusing on the pulmonary vascular remodeling. Bioactive lipids produced by perivascular inflammatory cells might modulate the vascular remodeling. Here, we show that ω-3 fatty acid-derived epoxides (ω-3 epoxides) released from mast cells by PAF-AH2, an oxidized phospholipid-selective phospholipase A2, negatively regulate pulmonary hypertension. Genetic deletion of *Pafah2* in mice accelerate vascular remodeling, resulting in exacerbation of hypoxic pulmonary hypertension. Treatment with ω-3 epoxides suppresses the lung fibroblast activation by inhibiting TGF-β signaling. In vivo ω-3 epoxides supplementation attenuates the progression of pulmonary hypertension in several animal models. Furthermore, whole-exome sequencing for patients with pulmonary arterial hypertension identifies two candidate pathogenic variants of *Pafah2*. Our findings support that the PAF-AH2-ω-3 epoxide production axis could be a promising therapeutic target for pulmonary hypertension.

[1] Department of Cardiology, Keio University School of Medicine, Tokyo, Japan. [2] Laboratory of Health Chemistry, Graduate School of Pharmaceutical Sciences, University of Tokyo, Tokyo, Japan. [3] Department of Biochemistry, Keio University School of Medicine, Tokyo, Japan. [4] Center for Medical Genetics, Keio University School of Medicine, Tokyo, Japan. ✉email: jinendo@keio.jp

Pulmonary arterial hypertension (PAH) is a rare, fatal disease that causes idiopathic pulmonary artery stenosis, which leads to increased pulmonary artery pressure and this chronic pressure-overload eventually results in right heart failure and death. Pulmonary hypertension (PH) is characterized by irreversible tissue changes, known as "pulmonary vascular remodeling", involving pulmonary artery endothelial cells, smooth muscle cells, and fibroblasts[1,2]. Although available therapies for PH have notably improved the survival of patients with PAH[3], a significant portion of patients do not achieve the expected efficacy. Therefore, medications that can suppress pulmonary vascular remodeling and reduce the disease progression are considered an unmet medical need that can potentially increase patient survival[4].

Inflammatory cells play an important role in tissue remodeling. In pulmonary vascular remodeling, several types of inflammatory cells present in the lung tissue produce humoral factors, such as cytokines and chemokines, that control the alterations of the vascular tissue[1,2,5]. Additionally, lipid mediators are also produced by local inflammatory cells and these mediators regulate inflammation, thrombus formation, angiogenesis and fibrosis, all of which can accelerate vascular remodeling[6,7]. In fact, it has been shown that proinflammatory functional lipids, such as prostanoids and leukotrienes, contribute to the pathogenesis of PH[8–10]. On the other hand, ω-3 fatty acids, primarily eicosapentaenoic acid (EPA) and docosahexaenoic acid (DHA), are known to play a bioprotective role, and some of their derivatives have been reported to possess unique functions, which can suppress tissue remodeling[6,11–13]. Using a pressure overload-induced cardiac remodeling model, our previous report demonstrated that EPA metabolites released by macrophages suppressed the abnormal activation of cardiac fibroblasts and maintained tissue homeostasis[14]. Therefore, ω-3 fatty acids and their derivatives are also expected to suppress pulmonary vascular remodeling in PH, but this remains to be determined.

Here, by a comprehensive lipidomic analysis of PH lung samples and phenotypic analysis of knock-out (KO) mice of type II platelet-activating factor acetylhydrolase (PAF-AH2), an ω-3 epoxide-producing enzyme from membrane phospholipids, we revealed that epoxidized ω-3 fatty acids have a 3-membered ring ether (ω-3 epoxides; 17,18-EpETE and 19,20-EpDPE) as functional lipid mediators involved in the pathogenesis of PH. Omega-3 epoxides were constantly produced from mast cells in the lung to suppress abnormal activation of adventitial fibroblasts, and they exhibited therapeutic effects on PH even when administered externally. We also found pathogenic mutations of PAF-AH2 in PAH patients who were insufficiently responsive to current therapeutics, suggesting that ω-3 epoxide might be a valuable therapeutic target for PAH.

## Results

**Omega-3 epoxides are reduced in the lungs of hypoxia-induced PH mice.** To identify valuable functional lipids that were characteristically altered in PH lungs, we performed comprehensive lipidomic analysis using liquid chromatography tandem mass spectrometry (LC-MS/MS) to determine the EPA, DHA, and arachidonic acid (AA) metabolites in the lung tissues of mice with PH induced by chronic hypoxia (10% oxygen concentration). We found that the levels of ω-3 epoxides, 17,18-EpETE and 19,20-EpDPE, and their inactive dihydrodiols, 17,18-diHETE and 19,20-diHDoPE, were reduced in tissues of hypoxic lungs (Fig. 1a–c, Supplementary Fig. 1a–e). Corresponding to these results, the expression of PAF-AH2, a key enzyme which belongs to the phospholipase A2 family and preferentially hydrolyzes ω-3 epoxide–containing membrane phospholipids to liberate ω-3

epoxides[15], was also significantly reduced in tissues of hypoxic lungs (Fig. 2a).

**Hypoxia-induced PH is exacerbated in Pafah2 KO mice.** In order to determine the significance of ω-3 epoxides in the development of PH, we investigated the phenotype of PAF-AH2-deficient mice subjected to chronic hypoxia (8 weeks). Pafah2 KO mice produced lower amounts of ω-3 epoxides, especially 17,18-EpETE and 19, 20-EpDPE, in their lungs under hypoxic conditions, compared to the wild type (WT) mice (Fig. 2b, Supplementary Fig. 2a). Histological analysis revealed that Pafah2 KO mice with PH showed advanced pulmonary vascular remodeling, such as thickening of the pulmonary artery wall and severe perivascular fibrosis (Fig. 2c, d). Consistent with these findings, Pafah2 KO mice had higher right ventricular systolic pressure (RVSP) during cardiac catheter examination than the WT mice, thereby indicating an increased severity of PH (Fig. 2e). Furthermore, Pafah2 KO mice with hypoxic PH exhibited advanced right heart failure, such as increased right ventricular (RV) hypertrophy (Fig. 2f) and higher mRNA levels of Nppa, a heart failure marker, in the RV (Fig. 2g), resulting in a high mortality rate of nearly 80% in 100 days after hypoxic exposure (Fig. 2h). Although we evaluated impacts of sex on PH mouse model, no differences were observed in the severity of hypoxic PH and in the phenotype of exacerbated PH in Pafah2 KO mice (Supplementary Fig. 3a–d). The hypoxia-induced pulmonary vascular remodeling of the Pafah2 KO mice was characterized by severe perivascular fibrosis, as indicated by the higher Col1a1 mRNA levels as well as the Vimentin-positive areas in the pulmonary vessels (Fig. 2i, j). Additionally, we evaluated the phenotype of PH using plasma-type PAF-AH KO mice (Pla2g7 KO), an enzyme that has PAF-acetyl hydrolase enzymatic activity similar to those of PAF-AH2[16,17], but no severe alterations in the PH were observed (Fig. 2c–f, h). These findings suggest that the lipid metabolites unique to PAF-AH2, for instance ω-3 epoxides, play a major role in the progression of PH.

**Mast cell-derived ω-3 epoxides suppress the progression of PH.** Next, we identified the type of cells responsible for producing ω-3 epoxides in the lungs. Fluorescent immunohistochemistry of lung samples from the hypoxic PH mouse model, Sugen/hypoxia PH mouse model, and human idiopathic PAH patients revealed that the cells expressing PAF-AH2 were tryptase-positive mast cells, but not macrophages, fibroblasts, endothelial cells, and respiratory epithelial cells (Fig. 3a, Supplementary Fig. 4a, b). Consistent with the findings of previous reports[18,19], mast cells were found to accumulate around pulmonary vessels in PH lungs (Fig. 3b).

To determine whether PAF-AH2 expressing mast cells contributed to the severe PH phenotype seen in Pafah2 KO mice, we prepared mast cell-deficient mice (Kit^(W-sh/W-sh)) and reconstituted them with bone marrow-derived mast cells (BMMCs) from WT mice or Pafah2 KO mice. We then exposed these mice to hypoxia for 4 weeks (Fig. 3c). At 4 weeks after the BMMC injection, a sufficient number of Tryptase-positive, PAF-AH2-positive mast cells could be confirmed in the lungs of the reconstituted mice (Supplementary Fig. 4c, d). As expected, Kit^(W-sh/W-sh) mice reconstituted with Pafah2 KO BMMCs exhibited a severe PH phenotype in comparison with those reconstituted with WT BMMCs (Fig. 3d–i), suggesting that the lack of PAF-AH2 activity in the mast cells was responsible for the phenotype of the Pafah2 KO mice under hypoxic conditions.

Omega-3 epoxides have been known to promote IgE-dependent degranulation in mast cells[15]. We investigated whether hypoxia-induced degranulation of mast cells was affected by the presence or absence of PAF-AH2 in vivo. While hypoxia

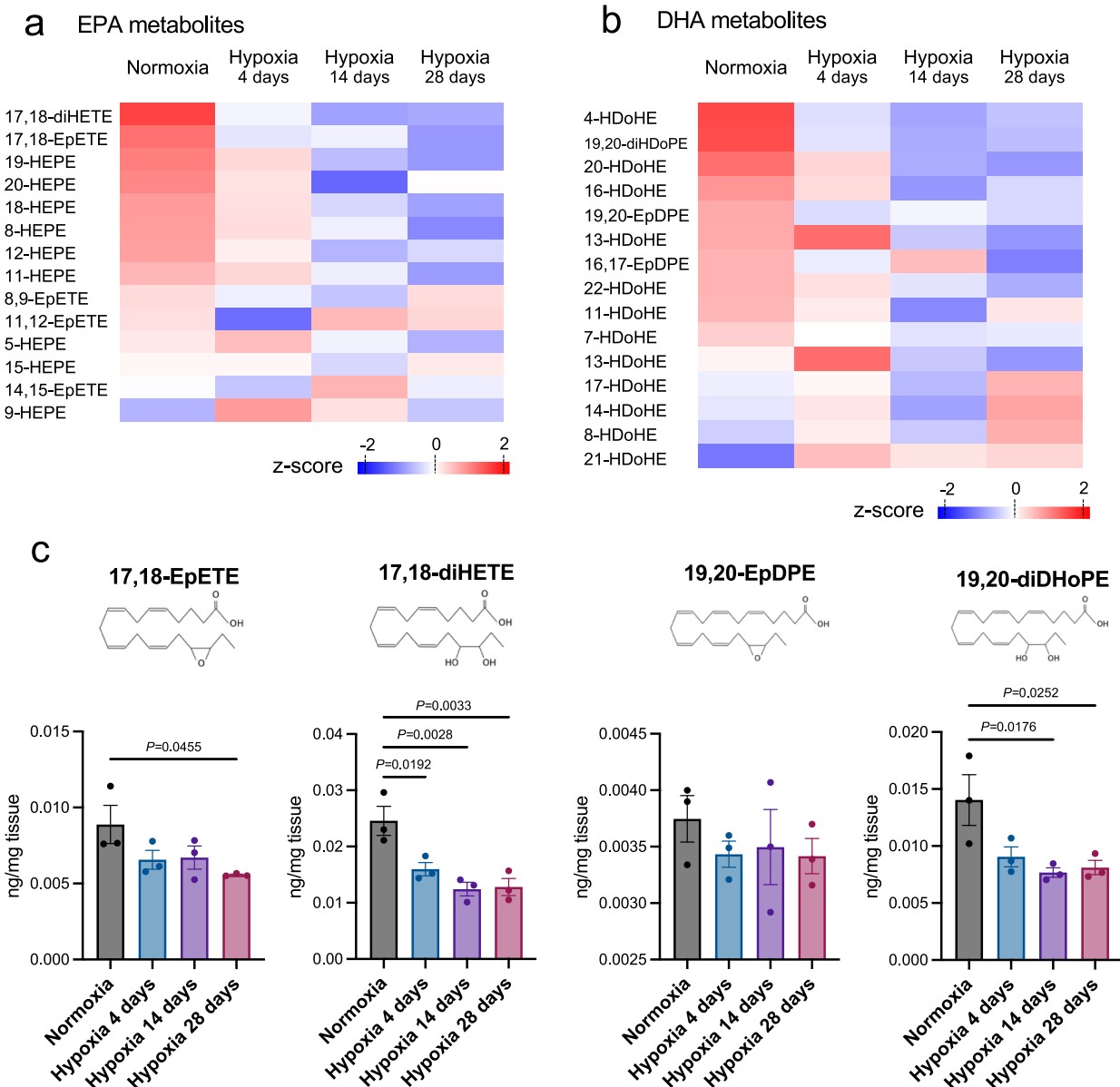

**Fig. 1 Omega-3 epoxides in murine lungs are reduced with exposure of hypoxia. a**, **b** Omega-3 fatty acid metabolites in LC-MS/MS based-lipidomics of the lungs of WT mice exposed to normoxia or hypoxia (10% $O_2$) for 4, 14, and 28 days. **a** EPA metabolites, **b** DHA metabolites. Z-score was calculated from the average value of each group ($n = 3$) and shown as heatmap. **c** The contents of ω-3 epoxides (17,18-EpETE and 19,20-EpDPE) and their dihydrodiols (17,18-diHETE and 19,20-diHDoPE) assessed by LC-MS/MS based-lipidomics of lungs of WT mice subjected to normoxia or hypoxia for 4, 14, and 28 days ($n = 3$). Data are mean ± SEM. P values were determined by one-way ANOVA with Dunnett's *post hoc* test.

increased the mast cells in the lungs and slightly augmented their degranulation, the total number and the degranulation rate of the mast cells in the lungs showed no significant difference between the WT mice and *Pafah2* KO mice under either normoxia or hypoxia (Supplementary Fig. 5a, b). Furthermore, hypoxia-dependent degranulation of BMMCs assessed by β-HEX release in vitro was nearly comparable between the WT BMMCs and *Pafah2* KO BMMCs (Supplementary Fig. 5c).

In order to investigate whether the degranulation of mast cells mediated the exacerbation of hypoxic PH in the absence of PAF-AH2 in vivo, we evaluated the improvement on hypoxic PH in *Pafah2* KO mice when administered with Ketotifen, a mast cell stabilizer which inhibits its degranulation. Although Ketotifen significantly suppressed hypoxia-dependent degranulation of the mast cells in the lungs (Supplementary Fig. 5d), the *Pafah2* KO mice exhibited severe PH regardless of Ketotifen treatment

(Supplementary Fig. 5e–g), indicating that degranulation of the mast cells was not involved in the mechanism underlying the severe PH induced by the absence of PAF-AH2.

**Omega-3 epoxides regulate PH by inhibiting fibroblast activation.** Since mast cells secrete ω-3 epoxides even under resting conditions, independent of IgE-antigen stimulation[15], we hypothesized that the ω-3 epoxides from mast cells acted on and controlled the constitutive cells of the pulmonary artery in a paracrine manner, especially perivascular fibroblasts which were significantly activated in hypoxia-exposed *Pafah2* KO mice (Fig. 2i, j). To examine the impact of the lipids released by mast cells on lung fibroblasts, we added lipid extracts of the free fatty acid fraction isolated from the culture supernatants of BMMCs to primary cultured murine lung fibroblasts. The lipid extracts from

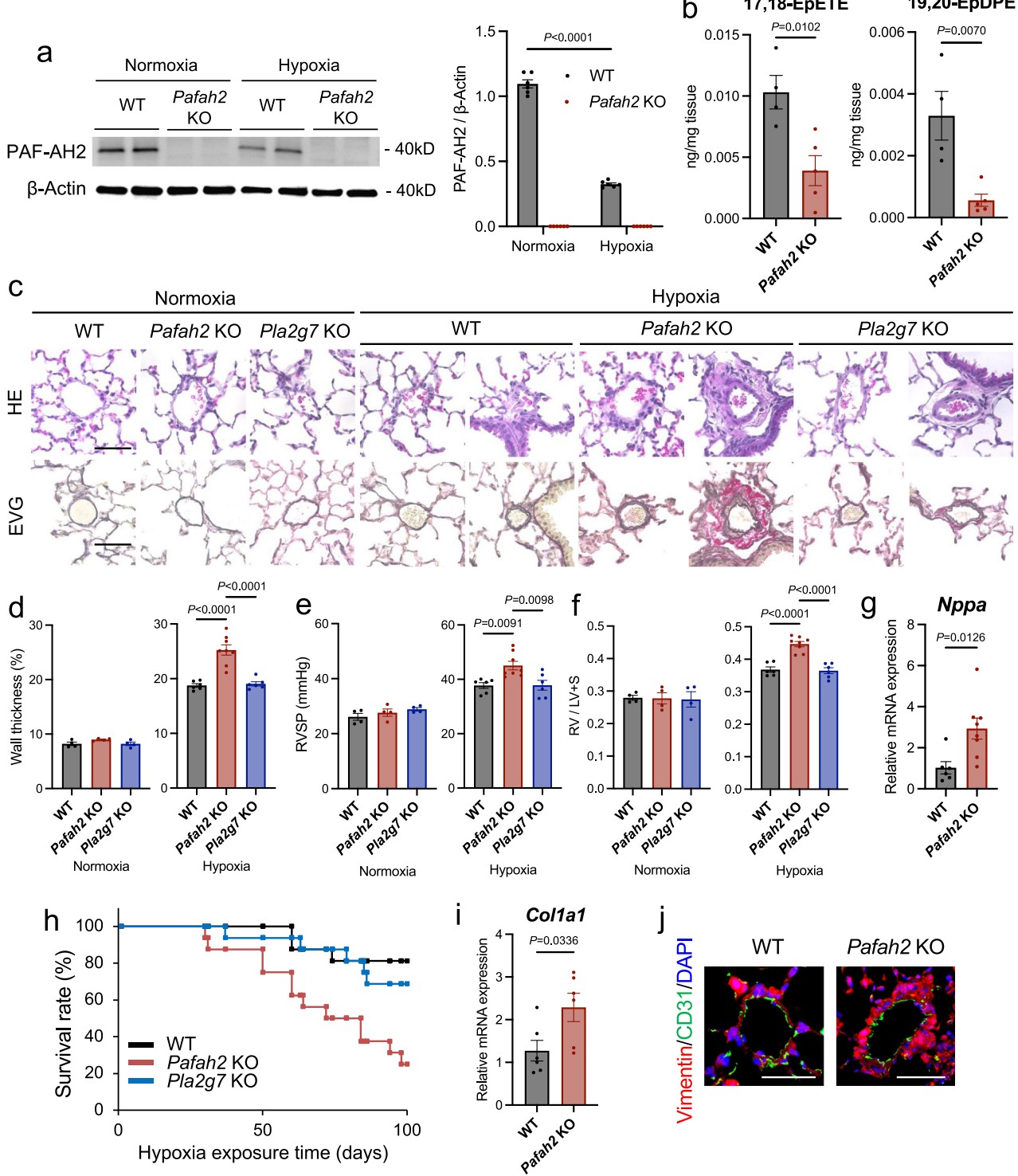

the *Pafah2* KO BMMCs increased the proliferation of fibroblasts and mRNA expressions of fibroblast activation markers, including *Col1a1* and *Acta2*, compared to the supernatants of the WT BMMCs. Additive treatment with ω-3 epoxides suppressed the aberrant proliferation and upregulation of the activation marker genes in the fibroblasts stimulated with lipid extracts from the *Pafah2* KO BMMCs (Fig. 4a, Supplementary Fig. 6a). The proliferation of lung fibroblasts was suppressed by treatment with ω-3 epoxides, 17,18-EpETE or 19,20-EpDPE, but not by EPA, DHA, or their dihydrodiols (Fig. 4b, Supplementary Fig. 6b). Focusing

on TGF-β signaling, which is closely related to tissue fibrosis and the pathophysiology of PAH[1,20], we evaluated the efficacy of ω-3 epoxides on lung fibroblasts. The treatment of ω-3 epoxides significantly suppressed the mRNA expression of *Col1a1, Acta2, and Il6* (Fig. 4c) and the expression of SM22α, a myofibroblast marker, in the lung fibroblasts under TGF-β-stimulated conditions (Fig. 4d), thereby indicating that ω-3 epoxides suppressed the activation of fibroblasts. Furthermore, ω-3 epoxides also exhibited an inhibitory effect on the migration of lung fibroblasts (Supplementary Fig. 6c). The other epoxidized fatty acid, an ω-6

**Fig. 2 Hypoxic PH is exacerbated in *Pafah2* KO mice. a** Western blotting of PAF-AH2 in total protein from the lungs of WT mice and *Pafah2* KO mice exposed to normoxia or hypoxia for 8 weeks (left) and quantification by densitometry (right). Experiments were repeated three times and the data were pooled. **b** The contents of ω-3 epoxides assessed by LC-MS/MS based-lipidomics of the **b** lungs in WT mice and *Pafah2* KO mice exposed to hypoxia for 8 weeks ($n = 4,5$). **c** Representative image of histological sections with HE staining (top) and EVG staining (bottom) of lungs in WT, *Pafah2* KO, and *Pla2g7* KO mice exposed to normoxia or hypoxia for 8 weeks. Scale bar, 50 µm. **d–f** The evaluation of PH severity in WT, *Pafah2* KO, and *Pla2g7* KO mice exposed to normoxia or hypoxia for 8 weeks (normoxia, $n = 4,4,4$; hypoxia, $n = 6,8,6$). Wall thickness of pulmonary arterioles (**d**), RVSP (**e**), weight ratio of RV to LV + septum (**f**). Experiments were repeated twice and the data were pooled (**e**, **f**). **g** Relative mRNA levels of *Nppa* in RV of hypoxia-exposed WT mice and *Pafah2* KO mice ($n = 6, 8$). Expression levels were normalized to those of 18S ribosomal RNA and then to those in the RV of WT mice. **h** Kaplan–Meier curves of hypoxia-exposed WT, *Pafah2* KO, and *Pla2g7* KO mice until 100 days ($n = 16$). Log-rank test; WT vs *Pafah2* KO, $P = 0.002$, *Pafah2* KO vs *Pla2g7* KO, $P = 0.011$. Experiments were repeated three times and the data were pooled. **i** Relative mRNA levels of *Col1a1* in lungs of WT mice and *Pafah2* KO mice exposed to normoxia or hypoxia for 8 weeks ($n = 6$). Expression levels were normalized to those of 18S ribosomal RNA and then to those in the lung of WT mice. **j** Representative image of Immunohistochemistry for Vimentin and CD31 in lungs of WT mice and *Pafah2* KO mice exposed to hypoxia for 8 weeks. Nuclei were labeled with DAPI. Scale bar, 50 µm. Data are mean ± SEM. *P* values were determined by two-way ANOVA with Bonferroni's *post hoc* test (**a**), one-way ANOVA with Tukey's *post hoc* test (**d–f**), or 2-tailed Student's *T* test (**b**, **g**, **i**).

epoxide including 14,15-EET, did not exhibit the inhibitory effects on TGF-β -activated lung fibroblasts (Supplementary Fig. 7a, b). It was confirmed that ω-3 epoxides inhibited the phosphorylation of Smad2, an upstream substrate of the TGF-β signaling pathway, suggesting this pathway as one of the mechanisms of action (Fig. 4e).

Previous studies have demonstrated that abnormalities in the endothelial cells and smooth muscle cells of the pulmonary artery are significantly involved in the pathogenesis of PAH[1,2]. However, ω-3 epoxides did not affect the gene expression associated with the pathophysiology of PAH in pulmonary artery endothelial cells (Supplementary Fig. 8a) and did not exert antioxidant effects on endothelial cells to protect against oxidative injuries (Supplementary Fig. 8b, c). Additionally, the proliferation of the pulmonary artery smooth muscle cells was not affected by treatment with the lipid extracts from BMMCs (Supplementary Fig. 8d) or with ω-3 epoxides (Supplementary Fig. 8e).

We also assessed a mechanism regulating the expression of PAF-AH2 in mast cells. As seen in hypoxic lungs in vivo (Fig. 2a), *Pafah2* expression was also downregulated in BMMCs when cultured under hypoxia in vitro (Supplementary Fig. 9a), whereas the expression of *Cyp4a12*, an enzyme responsible for ω-3 epoxidation[21], in BMMCs remained unchanged (Supplementary Fig. 9a). When administered with dimetyloxalyglycine (DMOG) or $CoCl_2$, activators of hypoxia inducible factor (HIF), the expression of *Pafah2* was reduced in BMMCs (Supplementary Fig. 9b), suggesting that PAF-AH2 was regulated by HIF.

**Administration of ω-3 epoxides improves PH in vivo.** We validated the therapeutic potential of ω-3 epoxides for PAH. In the chronic hypoxia PH mouse models, 19,20-EpDPE was administered two weeks after hypoxic exposure at a dose of 0.05 mg/kg/day by intra-peritoneal injection every day. 19,20-EpDPE administrations significantly improved PH by suppressing advanced pulmonary vascular remodeling including perivascular fibrosis both in the WT mice and in the *Pafah2* KO mice, but DHA or ω-6 epoxide, 14,15-EET, did not exhibit the beneficial effects (Fig. 5a–d, Supplementary Fig. 10a–d).

Additionally, we assessed the efficacy of ω-3 epoxides in a Sugen/hypoxia PH mouse model, which showed a more severe PH phenotype. Administration of 19,20-EpDPE, but not DHA, attenuated the severity of PH in the Sugen/hypoxia mice (Fig. 5e–h), as observed in the chronic hypoxic PH model. These results indicate that in vivo supplementation with ω-3 epoxides may be a viable treatment for PH.

**Pafah2 is a potent gene involved in the development of human PAH.** In order to investigate whether abnormalities of PAF-AH2 were involved in the development of human PAH, we searched for pathogenic variants in *Pafah2* among PH patients using whole-exome sequencing. We analyzed blood samples from 262 patients including 90 idiopathic PAH, 61 heritable PAH, and 54 connective tissue disease-associated PAH (Supplementary Table 1). We found two important variants of *Pafah2*, p.Arg85-Cys (R85C)/c.253C>T and p.Gln184Arg (Q184R)/c.551A>G, which were presumed to be highly pathogenic from three PAH patients (Supplementary Table 2). These SNPs had high Combined Annotation Dependent Depletion (CADD) scores that indicated high pathogenicity. Also, these variants were found at a site different from the catalytic sites of *Pafah2* (Fig. 6a). Using the homology model of PAF-AH2 proteins as the initial model, simulated PAF-AH2 p.R85C and PAF-AH2 p.Q184R variants were shown to have conformational changes compared to the native protein (Fig. 6b). Furthermore, we examined the impacts of the *Pafah2* variants by expressing the mutant proteins using pcDNA vectors in vitro. The levels of the mutant proteins, PAF-AH2 p.R85C and p.Q184R, were significantly reduced relative to those of the native PAF-AH2, but the level of PAF-AH2 S236C, a variant at the catalytic site, was unchanged (Fig. 6c). Interestingly, treatment with MG132, a proteasome inhibitor, partially recovered the protein levels of PAF-AH2 p.R85C and p.Q184R (Fig. 6c), suggesting that the protein degradation was due to the ubiquitin proteasome system. Taken together, the *Pafah2* p.R85C and p.Q184R variants found in PAH patients were considered to contribute to the progression of PH by enhancing the vulner-ability of the PAF-AH2 protein to degradation.

## Discussion

In this study, we demonstrated that ω-3 epoxides counteract the development of PH through regulating the vascular remodeling of pulmonary arteries. This is based on our findings that (a) ω-3 epoxides in the lung tissue decreased with the progression of PH, (b) PH was exacerbated in the mice lacking PAF-AH2, an ω-3 epoxide-producing enzyme, and (c) the supplementation of ω-3 epoxides attenuated the severity of PH in several disease models.

Epoxidized fatty acids with a 3-membered ring ether are highly reactive lipids; for instance, arachidonic acid-derived epoxidized fatty acids (epoxyeicosanoid; EET) have several functions that affect var-ious processes, such as angiogenesis and anti-inflammation, which contribute to the maintenance of homeostasis[22–24]. On the other hand, ω-3 fatty acids are known to have bioprotective effects including cardioprotection[25–27], and in recent years it has been revealed that their epoxy compounds exert strong unique physio-logical effects such as anti-inflammatory, vasodilatory, and tumor-suppressive effects[28–34]. Interestingly, the anti-fibrotic action and improvement of PH exerted by ω-3 epoxides observed in this study could not be confirmed when ω-3 fatty acids (EPA and DHA) were administered at the same dose, suggesting that these functions were

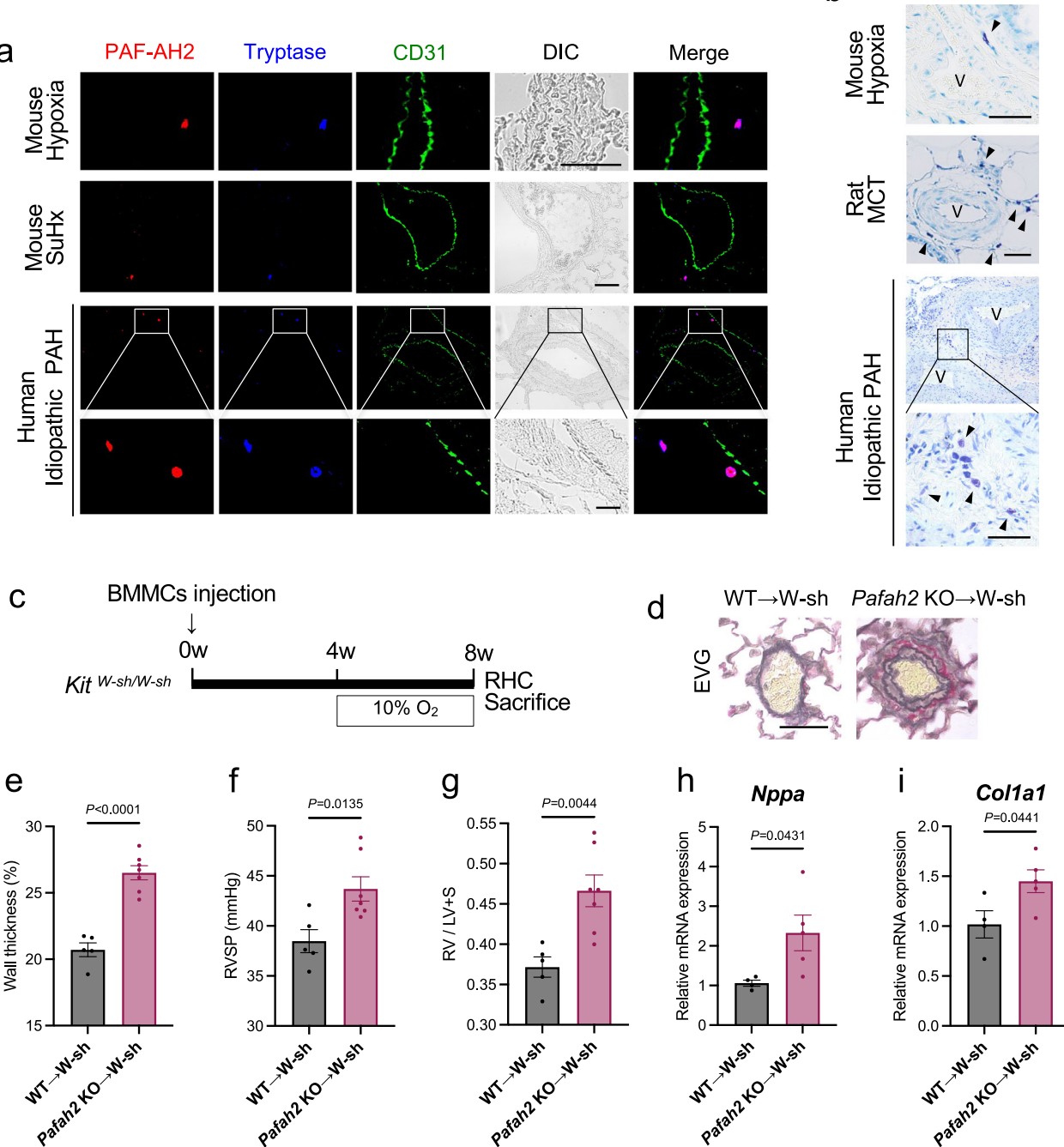

**Fig. 3 Mast cells are responsible for the phenotype of *Pafah2* KO mice. a** Representative image of immunostaining of PAF-AH2, Tryptase and CD31 in lung from PH mice or PAH patient. SuHx; Sugen/hypoxia. Scale bar, 50 μm. **b** Representative image of toluidine blue staining of lung sections in PH mice, PH rat, or PAH patient. Arrows indicate Toluidine blue-positive mast cells. V vascular lumen, MCT Monocrotaline. Scale bar, 50 μm. **c** Schematic diagram of *Kit* ^W-sh/W-sh^ mice reconstituted with BMMCs subjected to PH. RHC right heart catheterization. **d** Representative image of EVG staining of pulmonary arterioles in *Kit* ^W-sh/W-sh^ mice reconstituted with WT BMMCs (WT → W-sh) or *Pafah2* KO BMMCs (*Pafah2* KO → W-sh) after 4 weeks of hypoxic exposure. Scale bar, 50 μm. **e–g** The evaluation of PH severity of BMMC-reconstituted *Kit* ^W-sh/W-sh^ mice exposed to hypoxia (n = 5,7). Wall thickness of pulmonary arterioles (**e**), RVSP (**f**), weight ratio of RV to LV + septum (**g**). Experiments were repeated twice and the data were pooled (**f**, **g**). Relative mRNA levels of *Nppa* in RV (**h**) and *Col1a1* in lungs (**i**) of BMMC-reconstituted *Kit* ^W-sh/W-sh^ mice exposed to hypoxia. (n = 4,5). Expression levels were normalized to those of 18S ribosomal RNA and then to those in the RV or lung of WT → W-sh mice. Data are mean ± SEM. P values were determined by 2-tailed Student's T test.

specific to ω-3 epoxides. Although their points of action have not been definitively identified in detail thus far, the TGF-β signaling pathway may be involved in the underlying mechanism as 19, 20-EpDPE, an ω-3 epoxide, significantly suppressed the phosphorylation of Smad2, which is stimulated by TGF-β, in lung fibroblasts.

The substrate selectivity of plasma-type PAF-AH and PAF-AH2 is similar[16]. In addition to PAF, both PAF-AHs can hydrolyze phospholipids with short and/or oxidized sn-2 fatty acyl chain, but hardly hydrolyze phospholipids with two long fatty acyl chains. Recently it has been becoming clear that both

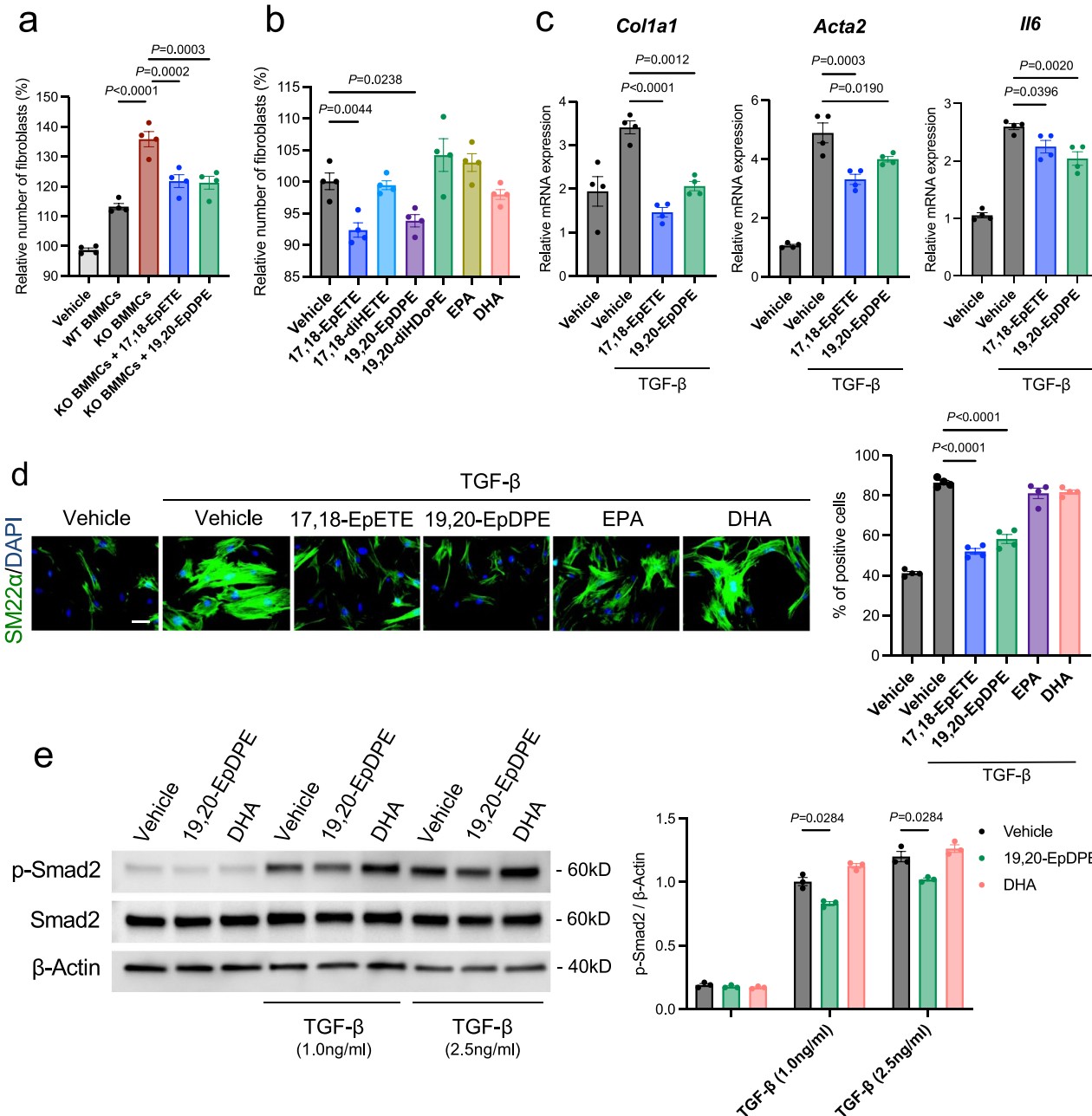

**Fig. 4 Omega-3 epoxides suppress the activation of lung fibroblasts. a** Relative number of lung fibroblasts stimulated by lipid extracts from cultured medium of BMMCs when treated with or without ω-3 epoxides (1 μM) for 48 h ($n = 4$). Data are representative of 2 independent experimental replicates. **b** Relative number of lung fibroblasts treated with vehicle, 17,18-EpETE (1 μM), 17,18-diHETE (1 μM), 19,20-EpDPE (1 μM), 19,20-diHDoPE (1 μM), EPA (1 μM), or DHA (1 μM) for 48 h ($n = 4$). Data are representative of 3 independent experimental replicates. **c** Relative expression levels of *Col1a1, Acta2*, and *Il6* mRNA in lung fibroblasts treated with vehicle, 17,18-EpETE (1 μM), or 19,20-EpDPE (1 μM) for 6 h when stimulated with or without TGF-β (2.5 ng/ml) ($n = 4$). Expression levels were normalized to those of 18S ribosomal RNA and then to those in the unstimulated control fibroblasts. Data are representative of 3 independent experimental replicates. **d** Immunostaining of SM22α in lung fibroblasts treated with vehicle, 17,18-EpETE (1 μM), 19,20-EpDPE (1 μM), EPA (1 μM), or DHA (1 μM) for 24 h when stimulated with or without TGF-β (2.5 ng/ml) (left). Scale bar, 50μm. Ratio of SM22α-positive cells to total lung fibroblasts (right) ($n = 4$). Data are representative of 2 independent experimental replicates. **e** Western blotting of pSmad2, Smad2, and β-Actin in total protein extracts from lung fibroblasts when administered vehicle, 17,18-EpETE (1 μM), or 19,20-EpDPE (1 μM) with or without TGF-β stimulation (1 ng/ml and 2.5 ng/ml) for 15 min (left) and quantification by densitometry (right). Experiments were repeated three times and the data were pooled. Data are mean ± SEM. *P* values were determined by one-way ANOVA with Dunnett's *post hoc* test (**a–d**) or two-way ANOVA with Tukey's *post hoc* test (**e**).

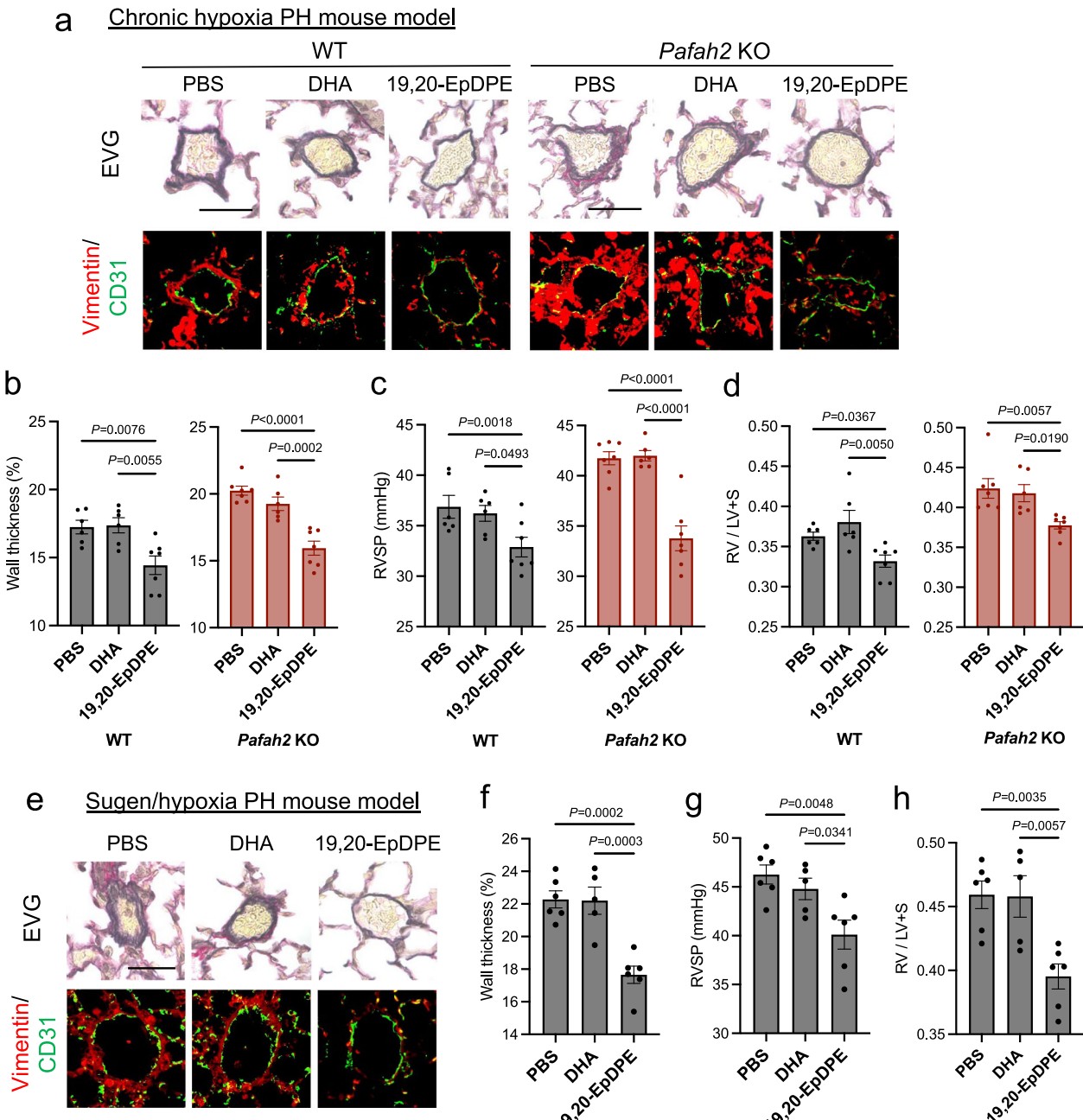

**Fig. 5 In vivo administration of ω-3 epoxides improves hypoxic PH and Sugen/hypoxia PH. a** Representative image of EVG staining (top) and Immunostaining of Vimentin and CD31 (bottom) in lungs of hypoxia-exposed WT mice and *Pafah2* KO mice when administered PBS, DHA (0.05 mg/kg/day), or 19,20-EpDPE (0.05 mg/kg/day) i.p. every day. These administrations were started 2 weeks after hypoxic exposure. Scale bar, 50 μm. **b–d** The evaluation of PH severity in hypoxia-exposed WT mice and *Pafah2* KO mice when administered PBS, DHA, or 19,20-EpDPE (WT mice, *n* = 6,6,7; *Pafah2* KO mice, *n* = 7,6,7). Wall thickness of pulmonary arterioles (**b**), RVSP (**c**), weight ratio of RV to LV + septum (**d**). Experiments were repeated twice and the data were pooled (**c**, **d**). **e** Representative image of EVG staining (top) and Immunostaining of Vimentin and CD31 (bottom) in lungs of Sugen/hypoxia treated WT mice when administered vehicle, DHA (0.05 mg/kg/day), or 19,20-EpDPE (0.05 mg/kg/day) i.p. every day. These administrations were started 3 weeks after hypoxic exposure. Scale bar, 50 μm. **f–h** The evaluation of PH severity in Sugen/hypoxia treated WT mice when administered PBS, DHA, or 19,20-EpDPE (*n* = 6,5,6). Wall thickness of pulmonary arterioles (**f**), RVSP (**g**), weight ratio of RV to LV + septum (**h**). Data are mean ± SEM. *P* values were determined by one-way ANOVA with Dunnett's *post hoc* test.

PAF-AHs can hydrolyzed non-fragmented oxidized phospholipids, such as F2-isoprostane-containing phospholipids, at a slow rate. Furthermore, plasma-type PAF-AH can also hydrolyze phospholipid hydroperoxides. On the other hand, it has been recently shown that PAF-AH2 has the unique activity that releases ω-3 epoxides from phospholipids[15]. We evaluated the severity of hypoxic PH in *Pla2g7* KO mice, but no significant

difference from WT mice was observed. Thus, we determined that the aggravated phenotype of hypoxic PH was specific to *Pafah2* KO mice and that the ω-3 epoxides produced by PAF-AH2 were closely related to the severity of PH. The role of plasma-type PAF-AH in hypoxic PH has never been reported and remains relatively unknown. In this study, the survival rate of *Pla2g7* KO mice with hypoxic PH is not significant but worse

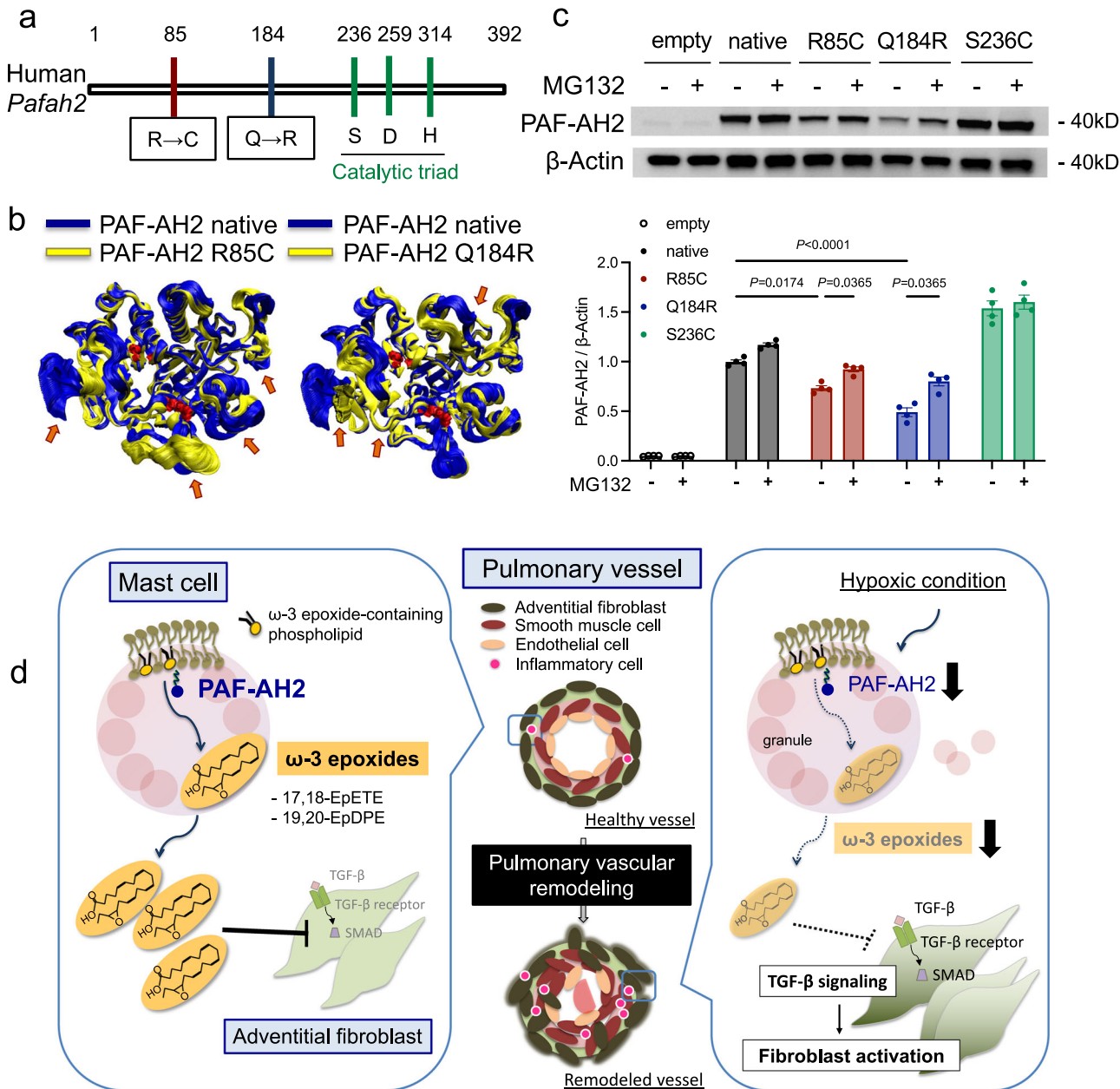

**Fig. 6 _Pafah2_ is a potent gene involved in the development of human PAH. a** Schematic diagram of the locations of 2 pathogenic mutation candidates in human _Pafah2_ gene. **b** The structural model of PAF-AH2 p.R85C variant (yellow in left), p.Q184R variant (yellow in right) and native form (blue), showing the stacked snapshots of 300 frames obtained from frame 2700–3000 (corresponding to 54–60 nano-seconds). Arrows show the changed conformation in each variant model compared to native form model. **c** Western blotting of PAF-AH2 in HEK 293 cells expressing human _Pafah2_ variants using pcDNA vectors when treated with or without MG132 (10 μM) for 6 h (top), and quantification by densitometry (bottom). Experiments were repeated four times and the data were pooled. **d** Graphical summary. Data are mean ± SEM. _P_ values were determined by two-way ANOVA with Tukey's _post hoc_ test.

than in control mice with hypoxic PH (Fig. 2h), suggesting that plasma-type PAF-AH might contribute to the vulnerability against hypoxic PH or hypoxia itself.

Cell type specificity also differs between the two enzymes. Plasma-type PAF-AH, which is secreted by monocytes, macrophages, and T lymphocytes, is predominantly expressed in the macrophage-rich area in atherosclerotic plaques[35]. In contrast, PAF-AH2 is expressed in hepatocytes, renal tubular epithelial cells, and highly expressed in mast cells[36]. Mast cells contain a considerable amount of PAF-AH2, suggesting that this enzyme is essential for the functions of mast cells. Shimanaka et al. have demonstrated that the ω-3 epoxides produced by PAF-AH2 promote degranulation in mast cells, which plays a major role in allergic inflammation[15]. On the other hand, our study showed that ω-3 epoxides from mast cells played a bioprotective role in the lungs when exposed to hypoxia. Additionally, Shimanaka et al. reported that ω-3 epoxides acted inside the mast cells[15]; while in our study, they were released from the cell to act on other surrounding cells. In fact, since ω-3 epoxides were detected in the culture supernatant of the BMMCs, it could be considered that mast cells released ω-3 epoxides to the extracellular space. Interestingly, mast cells have been known to slightly degranulate even when exposed to hypoxia[37], but no significant difference was observed in the WT BMMCs and the _Pafah2_ KO BMMCs, thereby indicating that ω-3 epoxides did not affect hypoxia-induced, IgE-independent degranulation.

Mast cells are widely present in the lungs as they are the major pulmonary inflammatory cells and are closely associated with allergic reactions such as asthma[38]. Mast cells have been reported to be involved in the pathogenesis of PH, and the degranulation of mast cells is thought to promote inflammation and exacerbate PH[18,19,39,40]. However, it is known that mast cells also possess anti-inflammatory properties[41,42]. Therefore, in studies using mast cell-deficient mice, it remains to be determined whether mast cells are beneficial or detrimental for PH. In our study, the administration of Ketotifen, a suppressor of degranulation of mast cells, did not affect the severity of hypoxic PH in mice. Therefore, under the conditions of this study, mast cells did not exacerbate the pathological state by degranulation but rather suppressed the progression of PH by releasing ω-3 epoxides.

Several animal models of PH have been developed in recent years, and limitations of each model should be comprehended. The PH model led by hypoxia alone may be better classified as group III PH (PH due to lung diseases and/or hypoxia) than group I PH (PAH). In this study, in order to clarify the relationship between a specific molecule and the pathophysiology of PH using genetically modified mice, we first used a single hit model with only hypoxic stimulation. The experimental results from the hypoxic PH model could partially explain the common mechanism underlying PH. However, the animal model in which the severity and tissue changes were similar to group I PH was also necessary to be used. To demonstrate the significance and effectiveness of the PAF-AH2-ω-3 epoxide axis in PAH, we conducted experiments in which ω-3 epoxides were administered to the Sugen/hypoxia PH mice in the present study.

Vascular remodeling in PH is mainly thought to be caused by abnormalities in vascular endothelial cells and smooth muscle cells. However, Pafah2 KO mice with hypoxic PH showed marked perivascular fibrosis, suggesting that the fibrosis strongly affected the exacerbation of PH. Interestingly, ω-3 epoxides did not significantly change gene expression and cell proliferation of pulmonary artery endothelial cells and smooth muscle cells in vitro. Additionally, mast cells releasing ω-3 epoxides into their surroundings were localized to the interstitial space around the blood vessels, which suggested that the perivascular fibroblasts were the target of the ω-3 epoxides.

In this study, we identified missense mutations that were closely associated with the pathogenicity of PH using a predictive program (CADD score >30; SIFT, deleterious; Polyphen; probably damaging) from whole-exome sequencing data in PAH patients. Most patients with these mutations were poorly responsive to existing therapeutic agents, primarily vasodilators. Additionally, experiments using the expression vector in cultured cells revealed reduced expressed protein level associated with the two mutations. The forced expression of PAF-AH2 in BMMC was attempted with the plasmid vector but was unsuccessful. Therefore, HEK293 cells, which are human-derived cells that produce a sufficient amount of target protein via transfection of plasmid vector and had low endogenous production of PAF-AH2 protein, were selected as transfected cells. Since treatment with a proteasome inhibitor restores the levels of the mutant proteins, we believe that post-translational modifications such as ubiquitination are involved in the reduction due to mutations. In future, a knock-in mouse of human PAF-AH2 harboring these mutations should be generated to determine if PH would naturally develop or show exacerbation.

The main treatment for PH includes functional vasodilation by various therapeutic agents, including prostaglandin I$_2$, nitric oxide, and phosphodiesterase-5 inhibition; however, there are no fundamental disease-modifying drugs that can suppress the progression of PH. Pulmonary artery remodeling, which involves inflammatory cells such as mast cells, is considered a key underlying mechanism in PH. In this study, we revealed that ω-3 epoxides produced by lung mast cells were candidates for an effective therapeutic drug that suppresses pulmonary vascular remodeling. In the future, we hope to develop a valuable therapeutic method using ω-3 epoxides for the purpose of suppressing pulmonary vascular remodeling in PAH patients with PAF-AH2 mutations who cannot be effectively treated with pulmonary artery dilators.

## Methods

**Study approval.** All procedures in the present study conformed to the principles outlined in the guide for the care and use of laboratory animals published by the US National Institutes of Health (NIH) and were approved by Laboratory Animal Center, Keio University School of Medicine (No. D2012-021 and No. 15063).

**Animals.** Pafah2 KO mice were backcrossed to C57BL/6 mice for over 10 generations[36]. Pla2g7 KO mice (C57BL/6J background) were kindly received from Dr Stafforini (the University of Utah). The male mice were used for all experiments except for those on Supplementary Fig. 3. In all experiments using KO mice (7–10 weeks), age and sex-matched C57BL/6J mice were used as controls. Mast cell-deficient Kit-mutant mice (C57BL/6J-Kit W-sh/W-sh) were purchased from the Jackson Laboratories. All mice were housed in climate-controlled (23 °C) specific pathogen-free facilities with a 12-h light-dark cycle, with free access to standard laboratory food (CE2; CLEA Japan Inc.) and water in Keio University.

**Materials.** EPA, DHA, AA, 17,18-EpETE, 19,20-EpDPE, 17,18-diHETE, 19,20-diHDoPE, 14,15-EET, and other fatty acid metabolites were obtained from Cayman Chemical.

**Lipid extraction from the lung samples.** Murine right lung tissues were harvested and immediately placed in liquid nitrogen. After freeze crushing of samples, lipids were extracted by the method of Bligh and Dyer[43]. The extracted solutions were dried up with centrifugal evaporator, dissolved in methanol: isopropanol = 1:1, and stored at −20 °C. Fatty acid metabolites were further purified from tissues by solid-phase extraction using InertSep NH2 columns (GL Science) with deuterium-labeled internal standard (11(12)-EET-d11). Briefly, InertSep NH2 columns were preconditioned with 6 ml of hexane and lipids extracted from tissues by the method of Bligh and Dyer were applied with 500 μL of chloroform. Columns were then washed with 6 ml of chloroform/isopropanol (2/1, v/v), followed by the elution with diethyl ether/acetic acid (98/2, v/v). The extracted solutions were dried up with centrifugal evaporator, dissolved in methanol: isopropanol = 1:1, and stored at −20 °C[44].

**Quantification of fatty acid metabolites (Lipidomic analysis).** For the detection of fatty acid metabolites, LC/ESI-MS-based lipidomics analyses were performed on a Shimadzu Nexera UPLC system (Shimadzu) coupled with a QTRAP 4500 hybrid triple quadrupole linear ion trap mass spectrometer (AB SCIEX). Chromatographic separation was performed on a ACQUITY UPLC HSS T3 column (2.1 × 100 mm, 1.8 μm; Waters) maintained at 40 °C using mobile phase A (water/acetic acid (100/0.1, v/v) containing 10 mM ammonium acetate) and mobile phase B (acetonitrile/methanol (4/1, v/v) containing 10 mM ammonium acetate) in a gradient program (0–2 min: 90% A; 2–10 min: 90% A → 30% A; 10–24 min: 30% A → 27% A; 24–27 min: 1% A; 27–32 min: 90% A) with a flow rate of 0.2 ml/min (0–10 min), 0.1 ml/min (10–15 min),0.2 ml/min (15–24 min) and 0.5 ml/min (24–32 min). The instrument parameters are as follows: curtain gas, 10 psi; ion spray voltage, −4500 V; temperature, 600 °C; ion source gas 1, 70 psi; ion source gas 2, 80 psi. The specific detection was performed by MRM as described previously[15,44].

**Experimental PH models.** In chronic hypoxia mice model, the mice were housed in a hypoxic chamber (10% O$_2$) maintained using a hypoxic air generator (TEIJIN) and monitored with an O$_2$ analyzer (JIKO-255), for 4 or 8 weeks. Sugen/hypoxia mice model was generated according to a previous report[45]. In brief, 20 mg/kg of VEGF inhibitor, Sugen (SU5416; S8442, Sigma-Aldrich), which was suspended in CMC (0.5% [w/v] carboxymethylcellulose sodium, 0.9% [w/v] sodium chloride, 0.4% [v/v] polysorbate 80, 0.9% [v/v] benzyl alcohol in deionized water), was subcutaneously injected into mice (C57BL/6J) at day 0, 7, and 14. The mice were housed in a hypoxic chamber (10% O$_2$) for 7 weeks. Animals were fed a standard diet. The studies were performed in accordance with National Institutes of Health Guidelines for the Care and Use of Laboratory Animals.

**Hemodynamic measurements.** Each mouse was anesthetized using 1.5% isoflurane on a heat board and monitored for heart rate and by electrocardiogram. A microtip catheter (Millar) was inserted into RV via the right jugular vein to measure RVSP. Hemodynamic measurements were analyzed using Lab Chart 8 (AD instruments). The heart was removed for assessment of RV hypertrophy [weight ratio of RV and (left ventricle + septum)] and RNA extraction, and the lungs were prepared for morphometric analysis and RNA extraction.

**Generation of mouse BMMCs**. BM cells obtained from mice were cultured in IL-3-containing BMMC complete medium comprising DMEM, 10% FBS, 2 mM L-glutamine, 100 IU/ml penicillin, 100 μg/ml streptomycin, 100 mM nonessential amino acids, and 5 ng/ml mouse rIL-3 to prepare BMMCs[46]. After 4–6 weeks of culture, >95% of the floating cells were confirmed to be Kit+ FcεRI+ mast cells by flow cytometry. Degranulation of BMMCs was evaluated by the amounts of released β-HEX in an enzymatic assay using 4-nitrophenyl N-acetyl-β-glucosaminide.

**Lipid extraction from the conditioned media of BMMCs**. To obtain conditioned media from BMMCs, $1 \times 10^7$ BMMCs were cultured in IL-3-containing BMMC complete medium for 2 days. Neutral lipids, phospholipids and free fatty acids were extracted from the conditioned media by solid-phase extraction using Sep-Pak C18 cartridges (Waters). Briefly, the conditioned medium was added to Sep-Pak columns and neural lipids were eluted with 10 ml of hexane. Free fatty acids were subsequently eluted with 10 ml of methyl formate. Phospholipids were finally eluted with 10 ml of methanol. The free fatty acid fraction was used as experiments.

**Adoptive transfer of BMMCs into mast-cell deficient mice**. BMMCs ($5 \times 10^6$ cells) were reconstituted by intravenous injection into 6-week-old male $Kit^{\text{W-sh/Wsh}}$ mice. Four weeks after reconstitution, mice were subjected to hypoxia to induce PH. The distribution and maturation of reconstituted BMMCs in lungs were evaluated by toluidine blue staining or fluorescent immunostaining.

**Inhibition of mast cell degranulation in vivo**. Ketotifen fumarate salt (Sigma) was dissolved in $H_2O$ and orally administered to mice for 4 weeks under hypoxic condition. The drinking water consumptions in mice with Ketotifen-administered group were not different from those with control group.

**Mast cell counting and evaluation of mast cell granulation**. Toluidine blue-positive mast cells were counted throughout lung section, which were measured at least 20 sections in each mouse under light microscope. Perivascular mast cells were categorized into granulated and degranulated based on the extrusion of secretory granules[18]. Granulated mast cells have dense cytoplasm, whereas degranulated mast cells have light cytoplasm with empty spots due to the discharge of secretory granules. An index of granulation was calculated as ratio of number of granulated mast cells/number of degranulated mast cells.

**Histological analysis and immunostaining**. The lungs of mice were harvested, inflated via the trachea with 4% paraformaldehyde at 25 $cmH_2O$ pressure, and then fixed in 4% paraformaldehyde. The samples were embedded in paraffin and sectioned at 4-μm thickness, and were stained with hematoxilin-eosin (HE), Elastica-van Gieson (EVG), and toluidine blue staining. At least 10 peripheral vessels (<100 μm diameter) exhibiting an approximately circulatory profile were randomly chosen in each lung section and analyzed the percent wall thickness using the following formula: [(vascular diameter − vascular lumen)/vascular diameter] × 100. All morphometric analyses were performed simultaneously and blinded to the study conditions.

For Immunohistochemistry of lung paraffin sections, retrieval of paraffin sections was performed using retrieval solution (Target retrieval solution S1700, Dako) in a boiling water bath for 20 min. The sections were cooled to room temperature for 30 min. After washing, the sections were incubated with the primary antibody (anti-CD31 [E-AB-70021; Elabscience; 1:200] for mouse samples or anti-CD31 [ab182981; Abcam; 1:2000] for human samples) at 4 °C overnight. The primary antibody was visualized using Alexa488-conjugated goat anti-rabbit IgG (Invitrogen; 1:2000). For counterstained with Vimentin, immunostaining was performed using the anti-Vimentin (ab8978; Abcam; 1:200) and M.O.M. fluorescein immunodetection kit with Avidin/Biotin Blocking kit (Vector Laboratories) and Streptavidin Texas Red (Vector Laboratories), according to the manufacturer's protocol. For counter stained with Tryptase and PAF-AH2, immunostaining was performed using the anti-Tryptase (NPB2-26444; Novus; 1:100) and M.O.M. fluorescein immunodetection kit with Avidin/Biotin Blocking kit and Streptavidin AMCA (Vector Laboratories). Anti-PAF-AH2 TI10 monoclonal antibody[36] (1:100) was labeled by HiLyte Fluor 555 Labeling kit (Dojindo Molecular Technologies). Sections were mounted with fluoromount-G with DAPI (Invitrogen) or Fluoromount (Diagnostic BioSystems). Images were acquired with a fluorescence microscope (BZ-9000; Keyence).

For "Supplementary Figures", immunostaining of murine lung paraffin sections was performed using the primary antibodies (anti-Podoplanin; AF3244; R&D Systems; 1:40, anti-Mac3; 550292; BD Biosciences; 1:100, and anti-Vimentin; ab92547; Abcam; 1:200) and visualized using Alexa 546-conjugated donkey anti-goat IgG, Alexa546-conjugated goat anti-rat IgG, or Alexa 594-conjugated goat anti-rabbit IgG (Invitrogen; 1:2000), respectively. For counterstained with PAF-AH2, immunostaining was performed using anti-PAF-AH2 TI10 monoclonal antibody (1:100) and M.O.M. fluorescein immunodetection kit with Avidin/Biotin Blocking kit and Fluorescein avidin DCS (Vector Laboratories). In immunostaining of human lung paraffin sections, the primary antibodies (anti-Podoplanin; AF3670; R&D Systems; 1:100, anti-CD68; #76437; Cell Signaling Technology; 1:200, and anti-Vimentin; ab92547; Abcam; 1:200) were used and visualized by Alexa594-conjugated donkey anti-sheep IgG or Alexa594-conjugated goat anti-rabbit IgG (Invitrogen; 1:2000), respectively. For counterstained with PAF-AH2, anti-PAF-AH2 TI10 antibody (1:100) and Alexa 488-conjugated goat anti-mouse IgG were used (1:2000).

**Immunocytochemistry**. Primary lung fibroblasts seeded at 35 mm glass base dish (Iwaki) were fixed with 4% paraformaldehyde and permeabilized with 0.2% Triton X-100 for 10 min, then blocked 1% serum for an hour. Cells were stained with primary antibodies overnight at 4 °C. Alexa488-conjugated goat anti-rabbit/mouse IgG (Invitrogen; 1:2000) was incubated for an hour at room temperature co-stained with DAPI. The primary antibodies used in the present study were anti-SM22α (ab14106; Abcam; 1:100) and anti-PCNA (ab29; Abcam; 1:100). Images were acquired with a fluorescence microscope (BZ-9000; Keyence).

**Isolation and culture of primary lung fibroblasts**. Lungs were perfused and digested with collagenase II (Worthington Biochemical Crop). Dissociated cells were incubated for an hour. The remaining cells were resuspended in DMEM (Wako) containing 10% fetal bovine serum and placed on Primaria culture dishes (BD). After 24 h, the culture medium was changed. After further 3 days and 5 days, the fibroblasts cultures were passaged, and used in the experiments.

**Cell proliferation assay**. Lung fibroblasts and smooth muscle cells were seeded at 96-well plates. After 12 h, ω-3 epoxides or lipid extracts were added to the plates, and cell proliferations were observed for 48 h using RealTime-Glo MT Cell Viability Assay (Promega) for 48 h on a microplate reader (Gen5 Synergy HTX; BioTek Instruments).

**Migration assay**. Cell migration was performed using a Boyden Chamber assay with 24 well, 8-μm pore size membrane invasion chambers (ThermoFisher). $2 \times 10^4$ cells were seeded into the upper chamber of the transwell. Omega-3 epoxides or ω-3 fatty acids were added to the lower chamber. After 24 h, membranes with migrated cells were fixed with methanol and stained with toluidine blue. Images were acquired with a microscope (BZ-9000; Keyence) and migrating cell numbers were quantified by 10× random fields.

**Cell culture**. Human pulmonary endothelial cells (Gibco) were cultured under 5% $CO_2$ at 37 °C using EGM-2 BulletKit medium (CC-3162; Lonza). Human pulmonary arterial smooth muscle cells (Kurabo) were cultured as well using DMEM (Wako) containing 20% fetal bovine serum medium. Cells were used between passages 4–8.

**Cellular reactive oxygen species assay**. Measurements of intercellular levels of reactive oxygen species were performed using Cellular ROS Assay Kit (abcam). Human pulmonary artery endothelial cells were seeded at 96-well plates. After 12 h, ω-3 epoxides or ω-3 fatty acids were added to the plates with or without $H_2O_2$ (500 μM or 1 mM) stimulation. An hour later, 2′,7′-dichlorofluorescin diacetate were added to the plates and 2′,7′-dichlorofluorescin were measured by fluorescence spectroscopy with excitation/emission at 485 nm/535 nm on a microplate reader (Gen5 Synergy HTX; BioTek Instruments).

**Measurements of pulmonary artery endothelial cell numbers**. Pulmonary artery endothelial cells were seeded at 96-well plates. After 12 h, ω-3 epoxides or ω-3 fatty acids were added to the plates with or without $H_2O_2$ (500 μM or 1 mM) stimulation. Two hours later, number of viable cells were measured using CellTiter-Glo Luminescent Cell Viability Assay (Promega) on a microplate reader (Gen5 Synergy HTX; BioTek Instruments).

**PAF-AH2 expression vectors and transfection into HEK293 cells**. We used the pcDNA3.1(+) plasmid vectors (V790202, Invitrogen) carrying a full length human Pafah2 cDNA (NCBI reference sequence: NM_000437.4) cloned from the human brain cDNA library (Life Technologies, Inc.)[47]. Using the Quik Change Site-Directed Mutagenesis Kit (Agilent) according to the instruction manual, we introduced single-base substitution to the plasmid carrying native Pafah2 cDNA to generate variants R85C (253C>T), Q184R (551A>G), and S236C (707C>G) and confirmed the presence of mutations by DNA sequencing. Native or mutant Pafah2 pcDNA plasmids (3.5 μg per 6-well dish) were transfected into HEK293 cells using Lipofectamine 2000 (Invitrogen). After 48 h, the culture medium was changed and exposed DMEM with or without MG132 (10 μM) for 6 h. Subsequently, cells were collected and the amount of expressed PAF-AH2 protein was analyzed by western blotting. HEK293 cells transfected by empty pcDNA vectors were used as controls.

**Western blot analysis**. Equal amounts of the total protein isolated from murine lungs, primary cultured fibroblasts, or HEK293 cells were prepared in radio-immunoprecipitation assay buffer containing 50 mM Tris–HCl (pH 7.6), 150 mM NaCl, 1% Nonidet-P40, 0.5% sodium deoxycholate, and 0.1% sodium dodecyl sulfate (SDS), and supplemented with 1 mM dithiothreitol (DTT), 100 nM MG132,

protease inhibitor cocktail, and phosphatase inhibitor cocktail (Nacalai Tesque). For western blot analysis, equal amounts of total protein (10–15 µg) from the lysate were subjected to SDS-polyacrylamide gel electrophoresis. The primary antibodies used in the present study were anti-PAF-AH2 (TI10; 1:1000), anti-β-Actin (sc-47778; Santa Cruz Biotechnology; 1:1000), anti-p-Smad 2 S465/467 (#3108), and anti-Smad 2 (#5339) (Cell Signaling Technology; 1:1000). Protein bands were visualized using horseradish peroxidase-conjugated secondary antibodies (1:2000) and enhanced chemiluminescence (Chemi-Lumi One Super; Nacalai Tesque) on a LAS-4000 mini (GE healthcare). Protein bands were quantified using ImageJ ver1.52.

**RNA extraction and real-time PCR**. For quantitative real-time PCR, total RNA samples from lungs, RV or lung fibroblasts were prepared using Trizol reagent (Invitrogen). Samples of total RNA (0.2–0.5 µg) were reverse-transcribed using High Capacity cDNA Reverse Transcription Kit (Applied Biosystems). Quantitative mRNA expression was assessed by real-time PCR using the THUNDERBIRD SYBR qPCR Mix (Toyobo). Samples were run on the ViiA7 (Applied Biosystems), and data were analyzed by the delta-delta CT method. The 18S ribosomal RNA was amplified and used as an internal control. The primer sequences for genes are listed in Supplementary Table S3.

**Whole-exome sequencing of human PH patients**. This study has been approved by the Institutional Review Board (IRB) of Keio University hospital (IRB No: 20140203), and all genetic tests were performed with informed consent from patients after genetic counseling. We included and analyzed blood samples from 262 patients including 90 idiopathic PAH, 61 heritable PAH, and 54 connective tissue disease-associated PAH (mean age, 45.5 ± 16.6 years; female, 78.0%; mean pulmonary artery pressure, 42.9 ± 15.5 mmHg). We performed whole-exome sequencing using a HiSeq 2500 platform (Illumina, San Diego, CA) and SureSelectXT Human All Exon Kit (Agilent Technologies, Santa Clara, CA) for hybridization capture. Pathogenicity of variants was assessed with CADD, SIFT, and PolyPhen-2[48,49]. Regarding to variants of *Pafah2* gene in the whole-exome sequencing data of patients with PH, we selected missense mutations with a CADD score of equal to or greater than 30 (MIS30). Known pulmonary arterial hypertension (PAH)-related genes includes bone morphogenetic protein receptor type 2 gene (*BMPR2*), activin A receptor–like 1 gene (*ACVRL1*), endoglin gene (*ENG*), caveolin-1 gene (*CAV-1*), T-box 4 gene (*TBX4*), potassium channel subfamily K member 3 gene (*KCNK3*), eukaryotic initiation translation factor 2 a kinase 4 (*EIF2AK4*), SMADs, adenosine triphosphate 13A3 (*ATP13A3*), aquaporin 1 gene (*AQP1*), growth differentiation factor 2 gene (*GDF2*), and SRY-related high-mobility group box family member 17 gene (*SOX17*), which were selected according to the previous report[48].

**Molecular dynamics simulation**. The initial 3D-structural model of PAF-AH2 was obtained using homology modeling using the SWISS-MODEL[50]. The crystal structure of plasma-type PAF-AH (PDB number: 5i9i.1.A) was used as template (the sequence identity 42.33%, the GMQE score 0.74)[51]. The molecular dynamics simulation was done on water-soluble condition using Nanoscale Molecular Dynamics (NAMD) version 2.12[52] with Chemistry at Harvard Molecular Mechanics (CHARMM) 36 force field[53]. The position and velocity vector of each atom and water molecule were calculated in each 2.0 femto-seconds ($1 \times 10^{-15}$ s) and Particle Mesh Ewald (PME) summation with a cut off length of 12 Å for the direct interactions was used for predicting long range electrostatic interaction[54]. The simulation was done with the options of rigibBonds all and the system was neutralized with NaCl in physiological condition (150 mEq/l). The water molecules were modeled as transferable intermolecular potential water molecules (TIP3P). The initial structure of each mutant was obtained by inducing the amino-acid substitution against the initial structure of wild-type in water-soluble condition using mutate residue plugin of Visual Molecular Dynamics (VMD) version 1.9.3[52]. The stable structure of each mutant along with the wild-type was calculated by running the simulation for 72 nano-seconds. Root mean square deviation (RMSD) was used to confirm that the calculation has stabilized (Supplementary Fig. 11a–c). The RMSD was calculated with distances relative to the initial model. While the initial model of the wild-type was the immediate output of the homology modeling solvated in water, the initial models for the mutants were the stable structure of the wild-type (after 60 nano-second simulation). All the results were visualized using VMD version 1.9.3. The simulation was done with the boundary size of 7.37 nm × 8.34 nm × 8.22 nm with a periodic boundary condition. The numbers of atom were 46628, 46615 and 46635 for WT, R85C and Q184R model (including waters and ions) respectively. The stacked snapshots of 300 frames obtained from frame 2700–3000 (corresponding to 54–60 nano-seconds), a time-window representing structural fluctuations after the model stabilized, were shown as figures.

**Statistics**. Data are presented as mean ± SEM. Z-score of lipidomics data was calculated using Microsoft Excel 2019 and shown as heatmap. The statistical significance of differences between the 2 groups was determined using unpaired Student's *T* test. Differences among multiple groups were compared using one-way or two-way ANOVA followed by post hoc tests. Statistical analyses were performed using GraphPad Prism version 9. A value of $P < 0.05$ was considered statistically significant.

**Reporting summary**. Further information on research design is available in the Nature Research Reporting Summary linked to this article.

## Data availability

The data supporting the findings from this study are available within the article and its supplementary information. The lipidomics data have been deposited in Metabolomics Workbench under accession number ST001951 (https://doi.org/10.21228/M8PH6J). The crystal structure of plasma-type PAF-AH used in this study are available in Protein Data Bank under PDB number: 5I9I (https://www.rcsb.org/structure/5I9I). The whole-exome sequencing data are not publicly available due to the presence of information that could compromise the privacy of research participants. In order to obtain the pseudonymized individual-level data, researchers need to contact the main member of the data access committee (M.K) at m.kataoka09@keio.jp and make scientifically appropriate requests. Overall, these data may only be used for research, and are not available for commercial use. All applications need to detail scientific purpose, objectives, methods, timetable, data management, ethical considerations, financial matters and competing interests, and also include a project plan and information about the entity responsible for the research as well as the principal investigator. All requests will be reviewed by the IRB of Keio University, and will require the requesting researcher to sign a data access agreement with Keio University. Data requests will be processed within 1 month if a written ethical approval is submitted to the author. The source data are provided with this paper.

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

## Acknowledgements

We would like to thank Yoshiko Miyake (Keio University) and Seiichi Kotoda (Bio research center company, Japan) for their excellent technical assistance. This work was supported by Japan Agency for Medical Research and Development (AMED) under grant number JP22gm5910014 (J.E.) and JP22gm1210013 (N.K.), SENSHIN Medical Research Foundation (J.E.), Japan Research Foundation for Clinical Pharmacology (J.E.), and Keio University Grant-in-Aid for Encouragement of Young Medical Scientists (H.M.).

## Author contributions

H.M. and J.E. designed and performed the experiments, analyzed the data and wrote the manuscript. M.K. and T.H. collected the patient samples and analyzed the data. Y.Shimanaka, Y.Sugiura and N.K. performed comprehensive lipidomic analysis. S.G. performed molecular dynamics simulation. H.K., N.Y., S.I., T.Y., K.S., A.A. and Y.K. performed the experiments and analyzed the data. K.K. performed whole-exsome sequencing. M.Suematsu, K.F., H.A. and M.Sano designed the experiments and interpreted the results.

## Competing interests

The authors declare no competing interests.
