## [Peer Review File · Nature Communications]

REVIEWER COMMENTS

Reviewer #1 (Remarks to the Author):

This study investigates the influences, as well as the mechanisms of action, of ω -3 epoxides and PAF-AH2 in mouse hypoxic PH and in clinical PAH. Employing a comprehensive lipidomic analysis, the authors first found that ω -3 fatty acid epoxides are decreased in a hypoxic mouse model of PH, which is accompanied by a reduction of the synthetic enzyme, PAF-AH2. Genetic deletion of Pafah2 gene exacerbates vascular remodeling and mouse PH. They located pulmonary mast cells to be the cellular sources PAF-AH2 and found that supplement of ω -3 epoxides attenuated the progression of PH. In an effort to demonstrate the molecular underpinning of bioprotective ω -3 epoxides, the authors showed that these mast cell-secreted lipids suppressed the activation of pulmonary fibroblasts through TGF- β inhibition. Whole-exome sequencing of patient PBMCs identified two pathogenic variants of Pafah2, prone to protein degradation. Overall, the idea that ω -3 epoxides/PAF-AH2 is protective from pulmonary vascular remodeling is novel; studies were well-designed and performed; data presentation is clear; and the evidence is convincing. Concerns and questions are provided below and, if appropriately addressed, would improve the quality and clarity of the paper:

1. The vimentin staining, shown in Figs 2j, 5a and 5e, should be performed with counter staining of α -SMA, CD31 or hyaluronidase to better illustrate the vascular structure. Specifically, the vimentin staining in 2j of Pafah2 KO appears to be somewhat non-specific, which makes it hard to conclude that the deleterious vascular effects of Pafah2 KO are concentrated on adventitial fibroblasts.
2. It will be helpful if the authors can expand the discussion of the differences between plasma PAF-AH and the type II enzyme (PAF-AH2), in terms of their substrate selectivity, cell type specificity, effects in cell apoptosis and differential roles in PH. Clearly, Pla2g7 KO also causes worse mouse survival in Fig 2h.
3. It is difficult to tell the proximity of the tryptase+, PAF-AH2 expressing mast cells to the PH vascular lesion in Fig 3a. Similarly, it is difficult to conclude the location of the toluidine blue+ cells in human PAH samples in Fig 3b.
4. Analysis of β -Actin needs to be included in Fig 4e to determine the suppression of SMAD2 activation.

5. The authors only used male mice in a disease which is predominantly female. Can they justify their use of males only. Did they look at female animals?

6. In recent efforts with modeling pulmonary vascular disease, there are an increasing number of animal models. Reliance of some of the work on hypoxia-induced changes alone is of unclear relevance to Group I PH (PAH). Hypoxia alone may more appropriately model Group III PH (not PAH). As the authors move between PH and PAH terminology as well as hypoxia (Group III) and Sugen/hypoxia (Group I PH?), it would be good to know what disease they are trying to model. We appreciate that animal modeling is always imperfect but it would be good to acknowledge this issue in some manner.

7. In the following publication, entitled "Mast cells promote lung vascular remodelling in pulmonary hypertension" (J. Hoffmann et al. European Respiratory Journal 2011 37: 1400-1410; DOI: 10.1183/09031936.00043310), the authors show that PH attributable to left heart disease was attenuated with the mast cell stabilizer ketotifen. Can the authors speculate on why this animal model was responsive but theirs was not?

Wen Tian MD

Mark Nicolls MD

Reviewer #2 (Remarks to the Author):

This is a well-written paper in which the authors show data indicating decreased levels of omega-3 fatty acid derived lipid mediators in the lung in hypoxia, as well as a protective effect of omega-3 derived epoxy metabolites in animal models of pulmonary hypertension. They base their focus on omega-3 epoxides as protective agents against pulmonary hypertension on the fact that they find an exacerbation of pulmonary hypertension in mice with a PAF-AH2 knockout.

While being a rather comprehensive study, there are several issues that this reviewer thinks need to be addressed before publication:

- In Figure 1, lipidomics data are shown only for EPA- and DHA-derived metabolites in hypoxia. It would be important to also show arachidonic acid- derived lipid mediators in this context and to compare the absolute amounts of these metabolites derived from these essential fatty acids.

- Also, it would be important to show, in the PAF-AH2 knockout, more data on the lipidome changes with hypoxia: While in Figure 2 data for two omega-3 epoxies are shown, there are no data on the other omega-3 epoxy metabolites, and also not on arachidonic acid derived epoxy metabolites (EETs) in this context: Could authors provide these data in order to clarify the specificity of the PAF-AH2 knockout with regard to formation of omega-3 epoxies?

- Furthermore, in order to establish specificity of omega-3 epoxy effects, did authors perform experiments using an arachidonic acid-derived epoxy compound (an EET) to compare findings between omega-3 and omega-6 derived epoxies in the experiments in Figure 4 and Figure 5?

Further questions:

- Maybe I missed this in the manuscript, but I did not find it in the methods: How were tissue samples prepared for lipidomics analyses? Given that PAF-AH2 catalyzes the release of omega-3 epoxies from phospholipids it would be important to only assess free lipid mediators and not include hydrolysis conditions leading to release of total preformed epoxy metabolites from the tissue phospholipids.

- The experiments were carried out exclusively with male mice. Is there a reason for this? – given that PH is more common in women...

- could authors give the gender ratio of the patients with PH they sequenced as part of this study? Interestingly, the 3 identified variants were all in female patients.

- also, could authors provide information on the percentage amount of the described deleterious PAF-AH2 mutants in the entire (healthy) population?

- In a small paragraph it is mentioned (bottom of page 10) that PAF-AH2 is possibly regulated in a HIF-dependent way. Do you have any ideas about the mechanism of this, since PAF-AH2 is down-regulated under hypoxia and HIF is classically a transcription factor under hypoxia but rather induces it?

- Regarding the experiments in which primary murine fibroblasts were cultured with supernatants of WT and Pafah2 KO BMMCs (Figure 4a). How were the concentration differences of 17,18-EpETE or 19,20-EpDPE between the lipid extracts from WT and KO BMMCs.

Reviewer #3 (Remarks to the Author):

The manuscript "Omega-3 fatty acid epoxides produced by PAF-AH2 in mast cells regulate pulmonary vascular remodeling" describes a series of experiments aimed at identifying the cause of pulmonary hypertension. Lipidomic analyses of lung samples, phenotypic analysis of knock-out mice, and tests with Omega-3 epoxides and PAF-AH2 support a role for w-3 fatty acids and their derivatives in vascular remodeling and pulmonary hypertension.

The work is generally well presented, but there are some aspects of the study that can be improved. Here I focus on the work done with PAF-AH2 in vitro and in silico.

1) Figure 6c shows data from experiments in which PAF-AH2 and variants were expressed in HEK293 cells using pcDNA vectors. There are two main problems with these data. First, it is unclear that HEK293 cells would provide relevant information about what happens with wild-type and mutant proteins in mast cells. HEK cells are not mast cells. The authors should at least acknowledge this limitation in the text. Second, information about the DNA constructs and methods used to express PAF-AH2 in HEK293 cells and to evaluate wild-type and mutant versions of PAF-AH2 are basically absent. There is no information on what exact vectors were used for expression in HEK 293 cells (were wild-type and mutants expressed using the same vectors? this is critical to know if quantification in Fig. 6c is meaningful), what exact wild-type sequence was used (NCBI accession code)? How were the mutations introduced and checked? and how did transfections and assessments of expression were done? (How many days after transfection were samples collected? Were the same DNA amounts used for all transfections?).

2) The section titled "Pafah2 is a potent gene involved in the development of human PAH" describes two variants that are presumed to be pathogenic. The text then focuses on molecular models and results from simulations that would suggest that mutants induce "morphological alterations compared to the native protein." I found this section poorly written and misleading. The authors should clarify that they are using homology models and apparently snapshots from simulations for their assessments. The term "estimation of the three-dimensional structures of ..." is unclear, and claiming that "the structures of the PAF-AH2 p.R85C and PAF-AH2 p.Q184R variants were shown to have morphological alterations ..." is incorrect: The work is presenting homology models or structural models at best, not structures. Similarly, the caption for panel b in figure 6 says "Three dimensional structures of ...", a misleading statement. Moreover, it is unclear how the comparison between wild-type and variant proteins was done using the simulation trajectories. Were just the final conformations compared? or were average conformations throughout trajectories compared? If so, why 72 ns would be enough? What metrics were used to determine differences? There are multiple methods that can be used to properly analyze the results from these simulations (clustering of conformations, RMSD and RMSF, etc.), but it is unclear whether any of these methods was used.

3) The "Molecular dynamics simulation" methods section describes how the initial model for PAF-AH2 model was obtained and simulated. I find the description insufficient. First, the statement "The initial 3D-structure of PAF-AH2 was obtained using homology modeling ..." is misleading. A structural model was obtained through homology modeling, in other words an experimentally based structure was not obtained as implied by using the word "structure". Second, there is little information about whether this model is good: What is the sequence similarity and identity between PAF-AH2 and the template used? Was the wild-type model stable during simulations (RMSD?). Second, a 2.0 fs time step was used, but it is unclear whether constraints were used for covalently bonded hydrogen atoms or not as required by such time step ("rigidBonds" option in NAMD to enable the shakeH algorithm). If this was not done, simulations need to be repeated with "rigidBonds all". Third, it is unclear whether the systems were neutralized (required for use of PME, and needed for wild-type *and* mutant systems to make sure that instability is not a PME issue). Fourth, it is unclear whether the simulations were done with physiological salt concentrations (and neutralized). Any co-factors or metal ions included? Last, size of each system (in nm and in number of atoms) is not stated, so we do not know if contacts with periodic images would be of concern (water box too small) and whether the wild-type and mutant systems are as similar as possible so as to eliminate biases in interpretation of results.

4) In the discussion, page 16, it is written that "the instability of the mutant proteins was clarified by expression in cultured cells using plasmid vectors." This sentence is misleading in multiple ways. First, this was done in HEK293 cells, which are not mast cells as stated above. Second, it is unclear that the same expression vectors were used (lack of detailed methods about these experiments as indicated above). Last, most structural biologists and biochemists would interpret "protein instability" in a very different way and would think of decreased melting temperature, folding issues, or conformational heterogeneity, none of which was really tested in this study. Also, mutants could be affecting catalytic activity allosterically, but this is not discussed. While these type of biochemical and biophysical experiments might be out of the scope of the present study, there should be clarity about the meaning of "instability" in this section and throughout the text.

General comments:

- Page 2. No need to define the KO abbreviation in the abstract if it is not used elsewhere in the abstract.

- Page 12. The sentence "Interestingly, treatment of MG132, ..." should probably be "Interestingly, treatment with MG132, ..."

- Page 13. The sentence "Interestingly, the anti-fibrotic action and improvement of PH exerted by ω -3 epoxides observed in this study could not be confirmed when ω -3 fatty acids were administered at the same dose, so it is suggested that ω -3 epoxides possessed the structure-specific functions." is very unclear. Not sure what study the authors are referring to and what they are suggesting.

- Page 14. The sentence "..., it could be considered that mast cells released ω -3 epoxides in an extracellular manner." Isn't this redundant? (released to the extracellular space?)

Reviewer #4 (Remarks to the Author):

Reviewer Critique

In a study, entitled "Omega-3 fatty acid epoxides produced by PAF-AH2 in mast cells regulate pulmonary vascular remodeling", the authors addressed the role of specific medications that suppress pulmonary vascular remodeling in pulmonary hypertension (PH), searching for functional lipids involved in the pathophysiology of the condition. The authors found that epoxidized ω -3 fatty acids (ω -3 epoxides) and the expression of PAF-AH2, a key enzyme that releases ω -3 epoxides from membrane phospholipids in mast cells, were reduced in the lungs of hypoxic PH mouse model. Pafah2 knock-out (KO) mice exhibited the exacerbation of hypoxic PH via advanced pulmonary vascular remodeling, whereas ω -3 epoxides suppressed the activation of lung fibroblasts through controlling TGF- β signaling. In vivo supplementation with ω -3 epoxides reduced the progression of PH not only in hypoxic exposure, but also during hypoxia. Analysis of whole-exome sequencing data from patients with PAH uncovered two candidate pathogenic variants of Pafah2 resulting in protein degradation. The authors conclude that their findings support that PAF-AH2/ ω -3 epoxide production axis could be a promising therapeutic target for PH. While this study provides potential new insights into the role of PAF-AH2 and ω -3 epoxide in PH of potential interest to the readership of Nature Communications, this is a candidate gene study with modest signal that needs to be replicated and further validated to ensure biological and statistical significance of the biological pathway(s) involved in a larger study. I have outlined my concerns in the comments below:

Major Comments:

1. The authors show modest reduction Omega-3 epoxides in murine lungs with exposure of hypoxia at 4, 14 and 28 days in n=3 compared to normoxia; how were these results controlled for; was any alternative method to that of LC-MS/MS used to validate these results.

2. The authors show that hypoxic pulmonary hypertension is exacerbated in Pafah2 KO mice. Did the authors control for %KO/KD values by sequencing and mRNA expression and is there a control gene expression. As shown in figure 2, several of the effects are modest and in the absence of proper control gene profile may be hard to interpret.

3. What control staining did the authors use to ensure cells other than mast cells are not contributing to the observed modest changes in phenotypes observed in Pafah2 KO mice and humans under hypoxic PH conditions; what about macrophages, fibroblasts, endothelial or epithelial cells.

4. The effects of omega-3 epoxides on suppression of cell activation and proliferation appears modest and vehicle alone follows same pattern for several of the measures; this should be presented in bar graph as % of control for better clarity.

5. The effects of ω -3 epoxides on improving hypoxic PH in mice/human similarly appear modest and unconvincing. This should be similarly presented as 5 control and ensure there is enough control comparison.

6. The potential role of Pafah2 in the development of human PAH is very preliminary and unclear about the pathogenic significance of these two variants despite positive scores and how many patients carry them. Are these variants listed as pathogenic in ClinVar, HGMD or other databases, how many subjects carry these variants, how many subjects are in gnomAD or other large public databases who carry these variants and may not have PH.

Minor comments:

1. In methods, should be separate section on human subjects describing their mutations and phenotypes and how these were validated.

REVIEWER COMMENTS

Reviewer #1 (Remarks to the Author):

This study investigates the influences, as well as the mechanisms of action, of ω -3 epoxides and PAF-AH2 in mouse hypoxic PH and in clinical PAH. Employing a comprehensive lipidomic analysis, the authors first found that ω -3 fatty acid epoxides are decreased in a hypoxic mouse model of PH, which is accompanied by a reduction of the synthetic enzyme, PAF-AH2. Genetic deletion of *Pafah2* gene exacerbates vascular remodeling and mouse PH. They located pulmonary mast cells to be the cellular sources PAF-AH2 and found that supplement of ω -3 epoxides attenuated the progression of PH. In an effort to demonstrate the molecular underpinning of bioprotective ω -3 epoxides, the authors showed that these mast cell-secreted lipids suppressed the activation of pulmonary fibroblasts through TGF- β inhibition. Whole-exome sequencing of patient PBMCs identified two pathogenic variants of *Pafah2*, prone to protein degradation. Overall, the idea that ω -3 epoxides/PAF-AH2 is protective from pulmonary vascular remodeling is novel; studies were well-designed and performed; data presentation is clear; and the evidence is convincing. Concerns and questions are provided below and, if appropriately addressed, would improve the quality and clarity of the paper:

We thank the reviewer for constructive comments. In the revised manuscript, we have included a substantial amount of new experimental data that should satisfactorily address the concerns of the reviewers.

1. The vimentin staining, shown in Figs 2j, 5a and 5e, should be performed with counter staining of α -SMA, CD31 or hyaluronidase to better illustrate the vascular structure. Specifically, the vimentin staining in 2j of *Pafah2* KO appears to be somewhat non-specific, which makes it hard to conclude that the deleterious vascular effects of *Pafah2* KO are concentrated on adventitial fibroblasts.

We thank the reviewer for the valuable suggestions. The endothelium was stained with CD31 in Figure. 2j, 5a, and 5e, to better illustrate the structure of blood vessels. For Figure 2j, we have replaced it with a clearer staining image for vimentin to better understand the proliferation of fibroblasts around the pulmonary arteries in *Pafah2* KO mouse.

2. It will be helpful if the authors can expand the discussion of the differences between plasma PAF-AH and the type II enzyme (PAF-AH2), in terms of their substrate selectivity, cell type specificity, effects in cell apoptosis and differential roles in PH. Clearly, *Pla2g7* KO also causes worse mouse survival in Fig 2h.

We thank the reviewer for the important suggestions. The following was added to the Discussion section in the manuscript.

Page 15, Line 3

“The substrate selectivity of plasma-type PAF-AH and PAF-AH2 is similar. In addition to PAF, both PAF-AHs can hydrolyze phospholipids with short and/or oxidized sn-2 fatty acyl chain, but hardly hydrolyze phospholipids with two long fatty acyl chains. Recently it has been becoming clear that both PAF-AH2 can hydrolyzed non-fragmented oxidized phospholipids, such as F2-isoprostane-containing phospholipids, at a slow rate. Furthermore, plasma-type PAF-AH can also hydrolyze phospholipid hydroperoxides. On the other hand, it has been recently shown that PAF-AH2 has the unique activity that releases ω -3 epoxides from phospholipids. We evaluated the severity of hypoxic PH in *Pla2g7* KO mice, but no significant difference from WT mice was observed. Thus, we determined that the aggravated phenotype of hypoxic PH was specific to *Pafah2* KO mice and that the ω -3 epoxides produced by PAF-AH2 were closely related to the severity of PH. The role of plasma-type PAF-AH in hypoxic PH has never been reported and remains relatively unknown. In this study, the survival rate of *Pla2g7* KO mice with hypoxic PH is not significant but worse than in control mice with hypoxic PH (Figure 2h), suggesting that plasma-type PAF-AH might contribute to the vulnerability against hypoxic PH or hypoxia itself.”

Regarding the impact on apoptosis, we examined the number of apoptotic cells around the pulmonary vessels by performing TUNEL staining in lungs of PH model mice. However, almost no TUNEL-positive cells were found in both *Pla2g7* KO mice and *Pafah2* KO mice. Therefore, we did not mention apoptosis in the manuscript.

3. It is difficult to tell the proximity of the tryptase+, PAF-AH2 expressing mast cells to the PH vascular lesion in Fig 3a. Similarly, it is difficult to conclude the location of the toluidine blue+ cells in human PAH samples in Fig 3b.

We thank the reviewer for the insightful comment. We have conducted immunostaining for CD31 in Figure 3a to visualize the morphology of pulmonary blood vessels and to easily capture the positional relationship with the tryptase⁺ PAF-AH2⁺ mast cells. In addition, the image in Figure 3b was replaced with an image containing an enlarged field of view so that the location of mast cells in vascular lesion in PH lung would be clearer, with the label "V" assigned to vascular lumens.

4. Analysis of b-Actin needs to be included in Fig 4e to determine the suppression of SMAD2 activation.

Thank you for highlighting this. We have performed western blotting for β -actin and the blot images were added to Figure 4e.

5. The authors only used male mice in a disease which is predominantly female. Can they justify their use of males only. Did they look at female animals?

We thank the reviewer for the important question. Although we evaluated the impact of sex on PH murine model, no differences due to sex were observed in the severity of hypoxic PH and in the phenotype of exacerbated PH in *Pafah2* KO mice (Supplementary Figure 3a-d). To prevent any unintended effects on study results, we have standardized sex of the mice in this study to male, as they are less susceptible to effects of the reproductive cycle. We have added the following to the Result section.

Page 6 , Line 10

“Although we evaluated impacts of sex on PH mouse model, no differences were observed in the severity of hypoxic PH and in the phenotype of exacerbated PH in *Pafah2* KO mice (Supplementary Figure 3a–3d).”

6. In recent efforts with modeling pulmonary vascular disease, there are an increasing number of animal models. Reliance of some of the work on hypoxia-induced changes alone is of unclear relevance to Group I PH (PAH). Hypoxia alone may more appropriately model Group III PH (not PAH). As the authors move between PH and PAH terminology as well as hypoxia (Group III) and Sugen/hypoxia (Group I PH?), it would be good to know what disease they are trying to model. We appreciate that animal modeling is always imperfect but it would be good to acknowledge this issue in some manner.

We thank the reviewer for the very constructive comments.

As highlighted and to avoid any confusion, differences between these PH models and the reason for their use were added to the Discussion as follows.

Page 17 , Line 8

“Several animal models of PH have been developed in recent years, and limitations of each model should be comprehended. The PH model led by hypoxia alone may be better classified as group III PH (PH due to lung diseases and/or hypoxia) than group I PH (PAH). In this study, in order to clarify the relationship between a specific molecule and the pathophysiology of PH using genetically modified mice, we first used a single hit model with only hypoxic stimulation. The experimental results from the hypoxic PH model could partially explain the common mechanism underlying PH. However, the animal model in which the severity and tissue changes were similar to group I PH was also necessary to be used. To demonstrate the significance and effectiveness of the PAF-AH2- ω -3 epoxide axis in PAH, we conducted experiments in which ω -3 epoxides were administered to the Sugen/hypoxia PH mice in the present study.”

7. In the following publication, entitled "Mast cells promote lung vascular remodelling in pulmonary hypertension " (J. Hoffmann et al. *European Respiratory Journal* 2011 37: 1400-1410; DOI: 10.1183/09031936.00043310), the authors show that PH attributable to left heart disease was attenuated with the mast cell stabilizer ketotifen. Can the authors speculate on why this animal model was responsive but theirs was not?

The previous report showed that mast cell degranulated in the pathogenesis of PH because ketotifen was effective in rats in which PH has been induced by left heart failure or by the administration of monocrotaline (Hoffmann, et al. *Eur Respir J* 2011; 37: 1400-1410, Dahal, et al. *Respir Res* 2011, 12:60). While in our hypoxic PH mouse model, ketotifen suppressed mast cell degranulation induced by hypoxic stimulation, but the degree of degranulation with or without PAF-AH2 were not significantly different. Furthermore, no change was observed in the severity of hypoxic PH even when treated with ketotifen at an appropriate dose. Therefore, these results suggested that degranulation itself contributed little to the pathological condition in the hypoxic PH mouse model.

The reason for the difference in these results remains unknown, but it could be largely due to the difference in the induction method of the PH model and animal species used. Hypoxia-induced degranulation is known to be mediated by TRP channels (Matsuda, et al. *J Clin Invest* 2017; 127(11): 3987-4000), but the mechanism of degranulation in other PH models is unknown. There may be differences in the degree and content of degranulation, or the response of the pulmonary artery to degranulation depending on the animal species, but no data are available. This would require further research.

Wen Tian MD

Mark Nicolls MD

Reviewer #2 (Remarks to the Author):

This is a well-written paper in which the authors show data indicating decreased levels of omega-3 fatty acid derived lipid mediators in the lung in hypoxia, as well as a protective effect of omega-3 derived epoxy metabolites in animal models of pulmonary hypertension. They base their focus on omega-3 epoxides as protective agents against pulmonary hypertension on the fact that they find an exacerbation of pulmonary hypertension in mice with a PAF-AH2 knockout.

We thank the reviewer for the positive comments. In the revised manuscript, we have included a substantial amount of new experimental data that should satisfactorily address the concerns of the reviewers.

While being a rather comprehensive study, there are several issues that this reviewer thinks need to be addressed before publication:

- In Figure 1, lipidomics data are shown only for EPA- and DHA-derived metabolites in hypoxia. It would be important to also show arachidonic acid- derived lipid mediators in this context and to compare the absolute amounts of these metabolites derived from these essential fatty acids.

We thank the reviewer for the constructive suggestions.

For arachidonic acid (AA) and AA-derived metabolites, the measurements that changed with hypoxic exposure were also shown in their absolute amounts (Supplementary Figure 1a–c). Regarding fatty acid as substrates, the amount of AA was higher than that of EPA or DHA, but for fatty acid metabolites, with a few exceptions, the amount of AA metabolites was not significantly different from that of EPA or DHA metabolites. EETs, AA-derived epoxides, also existed on a scale similar to that of omega-3 epoxies. In addition, the amount of substrates did not change significantly under hypoxic conditions. These findings suggest that the amount of fatty acid metabolites such as epoxides depend on the amount and activity of metabolic enzymes including PAF-AH2.

- Also, it would be important to show, in the PAF-AH2 knockout, more data on the lipidome changes with hypoxia: While in Figure 2 data for two omega-3 epoxies are shown, there are no data on the other omega-3 epoxy metabolites, and also not on arachidonic acid derived epoxy metabolites (EETs) in this context: Could authors provide these data in order to clarify the specificity of the PAF-AH2 knockout with regard to formation of omega-3 epoxies?

The measurements of the other epoxidized fatty acids are also shown in Supplementary Figure 2a. In either other ω -3 epoxides or EETs, no significant changes were observed between WT mice and *Pafah2* KO mice. These results were consistent with measurements of mast cell-derived fatty acid metabolites

previously reported by our colleague (*Nat Med* 2017, 23, 1287-1297), suggesting that PAF-AH2 could prefer to release ω -3 epoxides, especially 17,18-EpETE, and 19, 20-EpDPE.

- Furthermore, in order to establish specificity of omega-3 epoxy effects, did authors perform experiments using an arachidonic acid-derived epoxy compound (an EET) to compare findings between omega-3 and omega-6 derived epoxies in the experiments in Figure 4 and Figure 5?

We thank the reviewer for this important suggestion.

We investigated whether ω -6 epoxide (14,15-EET) can suppress the development of pulmonary artery remodeling in PH like ω -3 epoxides. First, in vitro, ω -6 epoxide was administered to TGF- β -activated lung fibroblasts, but it did not exhibit the inhibitory effects that were observed with ω -3 epoxide (Supplemental Figure 7a, b). Additionally, ω -6 epoxide did not suppress pulmonary vascular remodeling and PH in WT mice under hypoxic condition in vivo (Supplemental Figure 10a–d). These results demonstrated that the improvement in PH by the epoxidized fatty acid was not observed in ω -6 epoxide, and was specific to ω -3 epoxide.

We have added the following into the manuscript.

Page 10 , Line 3

“The other epoxidized fatty acid, an ω -6 epoxide including 14,15-EET, did not exhibit the inhibitory effects on TGF- β -activated lung fibroblasts (Supplemental Figure 7a, 7b).”

Page 11 , Line 11

“--, but DHA or ω -6 epoxide, 14,15-EET, did not exhibit the beneficial effects (Figure 5a-5d, Supplementary Figure 10a–10d).”

Further questions:

- Maybe I missed this in the manuscript, but I did not find it in the methods: How were tissue samples prepared for lipidomics analyses? Given that PAF-AH2 catalyzes the release of omega-3 epoxies from phospholipids it would be important to only assess free lipid mediators and not include hydrolysis conditions leading to release of total preformed epoxy metabolites from the tissue phospholipids.

Collected tissue sample was immediately frozen in liquid nitrogen, and thereafter the total lipid was extracted by the Bligh & Dyer method. The free fatty acid fraction was collected by solid-phase extraction with a column. There was no possibility of chemical hydrolysis due to esterification because of no basic conditions during extraction. As the reviewer has highlighted, the Methods section did not include how we have extracted lipids from the tissue samples, thus we revised it as follows:

Page 20, Line 15

“Lipid extraction from the lung samples

Details of the lipid extraction were previously described. Murine right lung tissues were harvested and immediately placed in liquid nitrogen. After freeze crushing of samples, lipids were extracted by the method of Bligh and Dyer. The extracted solutions were dried up with centrifugal evaporator, dissolved in methanol : isopropanol=1:1, and stored at -20°C . Fatty acid metabolites were further purified from tissues by solid-phase extraction using InertSep NH₂ columns (GL Science) with deuterium-labelled internal standard (11(12)-EET-d11). Briefly, InertSep NH₂ columns were preconditioned with 6 ml of hexane and lipids extracted from tissues by the method of Bligh and Dyer were applied with 500 μL of chloroform. Columns were then washed with 6ml of chloroform/isopropanol (2/1, v/v), followed by the elution with diethyl ether/acetic acid (98/2, v/v). The extracted solutions were dried up with centrifugal evaporator, dissolved in methanol : isopropanol=1:1, and stored at -20°C .”

- The experiments were carried out exclusively with male mice. Is there a reason for this? – given that PH is more common in women...

We thank the reviewer for the important question. Although we evaluated the impact of sex on PH murine model, no differences due to sex were observed in the severity of hypoxic PH and in the phenotype of exacerbated PH in *Pafah2* KO mice (Supplementary Figure 3a–d). To prevent any unintended effects on the study results, we have standardized the sex of the mice in this study to male, as they are less susceptible to the effects of the reproductive cycle. We have added the following to the Result section.

Page 6 , Line 10

“Although we evaluated the impacts of sex on PH mouse model, no differences were observed in the severity of hypoxic PH and in the phenotype of exacerbated PH in *Pafah2* KO mice (Supplementary Figure 3a-3d).”

- could authors give the gender ratio of the patients with PH they sequenced as part of this study? Interestingly, the 3 identified variants were all in female patients.

The gender ratio of the PH patients who underwent whole-exome sequencing in our study was 78.0 % for females and 22.0 % for males. The age and the mean pulmonary artery pressure were 45.5 ± 16.1 years (44 [33, 58] in the quartile) and 42.9 ± 15.5 mmHg (42 [31, 53] mmHg in the quartile), respectively. We have added the information in the Method section as below:

“-- (mean age, 45.5±16.6 years; female, 78.0%; mean pulmonary artery pressure, 42.9±15.5 mmHg)”

- also, could authors provide information on the percentage amount of the described deleterious PAF-AH2 mutants in the entire (healthy) population?

The percent of allele frequencies of these mutations in the global healthy population obtained from the gnomAD v2.1.1 (The Genome Aggregation Database) and the international large-scale gene database including a wide variety of the population, is 0.22 % (c.253C> T) and 0.0863 % (c.551A> G), respectively. The percentage of allele frequencies of each mutation in the Japanese general population obtained from ToMMo (Tohoku Medical Megabank Organization) 103.5KJSNV ver.1/2 data is also extremely low at 0.45 % (c.253C> T) and 0 % (c.551A> G). We have summarized the information in Supplementary Table 2.

- In a small paragraph it is mentioned (bottom of page 10) that PAF-AH2 is possibly regulated in a HIF-dependent way. Do you have any ideas about the mechanism of this, since PAF-AH2 is down-regulated under hypoxia and HIF is classically a transcription factor under hypoxia but rather induces it?

As the reviewer has highlighted, when HIF regulates directly as a transcription factor, the expression of downstream genes is generally enhanced. In contrast, it is well known that the expression of several genes are suppressed in a HIF-dependent manner, but the mechanism has only been partially elucidated. In the present study, we observed that the expression of PAF-AH2 was suppressed in a HIF-dependent manner. We performed a luciferase reporter assay to assess the activity of the promoter region of the *Pafah2* gene that contains 500bp fragment upstream from the transcription start site, but the transcriptional activity was not changed even when co-transfected with the plasmid vector of HIF-1α or treated with DMOG additionally, suggesting that PAF-AH2 is not directly regulated by HIF .

As a mechanism for downregulating gene expression in a HIF-dependent manner, it is considered that there is a possibility of indirect regulation mediated by transcription repressors such as DEC1 or

non-coding RNAs such as siRNA or lncRNA (*Nucleic Acids Res* 2010;38:2332-2345, *J Hepatol* 2012;56:41–447). In this study, the key molecule that suppresses the expression of PAF-AH2 has not been identified, and future research might be required to investigate it.

- Regarding the experiments in which primary murine fibroblasts were cultured with supernatants of WT and *Pafah2* KO BMMCs (Figure 4a). How were the concentration differences of 17,18-EpETE or 19,20-EpDPE between the lipid extracts from WT and KO BMMCs.

In this study, BMMCs were prepared under the same conditions as the previous study reported by our colleague (*Nat Med* 2017, 23, 1287-1297). The concentration of ω -3 epoxides in the supernatant was significantly less in *Pafah2* KO BMMCs than in wild type BMMCs (the concentration of 17,18-EpETE was 30 nM in WT vs 6 nM in KO, and that of 19,20-EpDPE was 45 nM in WT vs 10 nM in KO). In the experiment where fibroblasts were stimulated with the lipid extract of BMMC culture supernatant, we used the ω -3 epoxide in the lipid extract concentrated 10-fold during the extraction process. The ω -3 epoxide compound added to the fibroblasts stimulated with the lipid extract was used at 1 μ M based on the results of preliminary experiments, which is similar to the amount of ω -3 epoxide reduced by *Pafah2* KO.

Reviewer #3 (Remarks to the Author):

The manuscript "Omega-3 fatty acid epoxides produced by PAF-AH2 in mast cells regulate pulmonary vascular remodeling" describes a series of experiments aimed at identifying the cause of pulmonary hypertension. Lipidomic analyses of lung samples, phenotypic analysis of knock-out mice, and tests with Omega-3 epoxides and PAF-AH2 support a role for w-3 fatty acids and their derivatives in vascular remodeling and pulmonary hypertension.

The work is generally well presented, but there are some aspects of the study that can be improved. Here I focus on the work done with PAF-AH2 in vitro and in silico.

1) Figure 6c shows data from experiments in which PAF-AH2 and variants were expressed in HEK293 cells using pcDNA vectors. There are two main problems with these data. First, it is unclear that HEK293 cells would provide relevant information about what happens with wild-type and mutant proteins in mast cells. HEK cells are not mast cells. The authors should at least acknowledge this limitation in the text. Second, information about the DNA constructs and methods used to express PAF-AH2 in HEK293 cells and to evaluate wild-type and mutant versions of PAF-AH2 are basically absent. There is no information on what exact vectors were used for expression in HEK 293 cells (were wild-type and mutants expressed using the same vectors? this is critical to know if quantification in Fig. 6c is meaningful), what exact wild-type sequence was used (NCBI accession code)? How were the mutations introduced and checked? and how did transfections and assessments of expression were done? (How many days after transfection were samples collected? Were the same DNA amounts used for all transfections?).

We thank the reviewer for the valuable suggestions.

Regarding point 1, the following have been added to the Discussion section.

Page 18, Line 14

“The forced expression of PAF-AH2 in BMDC was attempted with the plasmid vector but was unsuccessful. Therefore, HEK293 cells, which are human-derived cells that produced a sufficient amount of target protein via transfection of plasmid vector and had low endogenous production of PAF-AH2 protein, were selected as transfected cells.”

We apologize that the explanation about the plasmid vector was missing in the text. We have added the following to the Methods section.

Page 26, Line 18

“PAF-AH2 expression vectors and transfection into HEK293 cells

We used the pcDNA3.1(+) plasmid vectors (V790202, Invitrogen) carrying a full length human *Pafah2* cDNA (NCBI reference sequence: NM_000437.4) cloned from the human brain cDNA library (Life Technologies, Inc.). Using the Quik Change Site-Directed Mutagenesis Kit (Agilent) according to the instruction manual, we introduced single-base substitution into the plasmid carrying native *Pafah2* cDNA to generate variants R85C (253C>T), Q184R (551A>G), and S236C (707C>G) and confirmed the presence of mutations by DNA sequencing. Native or mutant *Pafah2* pcDNA plasmids (3.5 µg per 6-well dish) were transfected into HEK293 cells using Lipofectamine 2000 (Invitrogen). After 48 hours, the culture medium was changed and exposed DMEM with or without MG132 (10µM) for 6 hours. Subsequently, cells were collected and the amount of expressed PAF-AH2 protein was analyzed by western blotting. HEK293 cells transfected by empty pcDNA vectors were used as controls.”

2) The section titled "Pafah2 is a potent gene involved in the development of human PAH" describes two variants that are presumed to be pathogenic. The text then focuses on molecular models and results from simulations that would suggest that mutants induce "morphological alterations compared to the native protein." I found this section poorly written and misleading. The authors should clarify that they are using homology models and apparently snapshots from simulations for their assessments. The term "estimation of the three-dimensional structures of ..." is unclear, and claiming that "the structures of the PAF-AH2 p.R85C and PAF-AH2 p.Q184R variants were shown to have morphological alterations ..." is incorrect: The work is presenting homology models or structural models at best, not structures. Similarly, the caption for panel b in figure 6 says "Three dimensional structures of ...", a misleading statement. Moreover, it is unclear how the comparison between wild-type and variant proteins was done using the simulation trajectories. Were just the final conformations compared? or were average conformations throughout trajectories compared?

We thank the reviewer for this important point.

We agree that this section the reviewer has highlighted was poorly written and misleading. In the revised manuscript, we have clarified that we were using a homology model rather than the structure obtained from crystallography as the initial model.

Page 12, Line 10

“Using the homology model of the PAF-AH2 proteins as the initial model, the simulated PAF-AH2 p.R85C and PAF-AH2 p.Q184R variants were shown to have conformational changes compared to the native protein (Figure 6b).”

Also, in the caption of Figure 6b, we have clarified that the figure is showing the stacked snapshots of 300 frames obtained from frame 2700–3000 (corresponding to 54-60 microseconds). Thus, the figure shows the “range” of conformational changes rather than capturing the single timepoint. This approach can

capture conformational differences of the 2 models that exceed the heat fluctuations.

The caption for Figure 6b.

“(b) The structural model of PAF-AH2 p.R85C variant (yellow in left), p.Q184R variant (yellow in right) and native form (blue), showing the stacked snapshots of 300 frames obtained from frame 2700–3000 (corresponding to 54–60 micro-seconds). Arrows show the changed conformation in each variant model compared to native form model.”

If so, why 72 ns would be enough? What metrics were used to determine differences? There are multiple methods that can be used to properly analyze the results from these simulations (clustering of conformations, RMSD and RMSF, etc.), but it is unclear whether any of these methods was used.

We used RMSD to confirm that the calculation has stabilized. We have also added figures for the RMSD plot showing stabilization of the model structure in 60 nano seconds. We have added these to the methods and figure legends as follows.

Page 30, Line 10

“Root mean square deviation (RMSD) was used to confirm that the calculation has stabilized (Supplementary Figure 11a-11c).”

3) The "Molecular dynamics simulation" methods section describes how the initial model for PAF-AH2 model was obtained and simulated. I find the description insufficient. First, the statement "The initial 3D-structure of PAF-AH2 was obtained using homology modeling ..." is misleading. A structural model was obtained through homology modeling, in other words an experimentally based structure was not obtained as implied by using the word "structure".

Second, there is little information about whether this model is good: What is the sequence similarity and identity between PAF-AH2 and the template used? Was the wild-type model stable during simulations (RMSD?).

We thank the reviewer for the constructive suggestions.

In the revised manuscript, we have clarified that the simulation was done on homology modeling and have made revisions. In addition, we have added the explanation on the quality of the homology model as follows.

Page 29, Line 15

" The initial 3D-structural model of PAF-AH2 was obtained using homology modeling using the SWISS-MODEL. The crystal structure of plasma-type PAF-AH (PDB number: 5i9i.1.A) was used as template (the sequence identity 42.33 %, The GMQE score 0.74)."

Second, a 2.0 fs time step was used, but it is unclear whether constraints were used for covalently bonded hydrogen atoms or not as required by such time step ("rigidBonds" option in NAMD to enable the shakeH algorithm). If this was not done, simulations need to be repeated with "rigidBonds all". Third, it is unclear whether the systems were neutralized (required for use of PME, and needed for wild-type *and* mutant systems to make sure that instability is not a PME issue). Fourth, it is unclear whether the simulations were done with physiological salt concentrations (and neutralized). Any co-factors or metal ions included?

The previous simulation was performed with the option of "rigidBonds all" and the system was neutralized with NaCl. However, the overall number of ions was very small and was not in physiological condition. Thus, we have re-ran the simulation in physiological salt conditions (150 mEq/l NaCl, neutralized, rigidBond all) according to the recommendation.

We have also added the following to the manuscript.

Page 30, Line 3

"The simulation was done with the options of rigidBonds all and the system was neutralized with NaCl in physiological condition (150 mEq/l)."

Last, size of each system (in nm and in number of atoms) is not stated, so we do not know if contacts with periodic images would be of concern (water box too small) and whether the wild-type and mutant systems are as similar as possible so as to eliminate biases in interpretation of results.

In the Methods section, we have stated the boundary size of each system in nm and in the number of atoms as follows.

Page 30, Line 12

"The simulation was done with the boundary size of 7.37 nm x 8.34 nm x 8.22 nm with a periodic boundary condition. The numbers of atom were 46628, 46615 and 46635 for WT, R85C and Q184R model (including waters and ions) respectively. "

4) In the discussion, page 16, it is written that "the instability of the mutant proteins was clarified by expression in cultured cells using plasmid vectors." This sentence is misleading in multiple ways. First, this was done in HEK293 cells, which are not mast cells as stated above. Second, it is unclear that the same expression vectors were used (lack of detailed methods about these experiments as indicated above). Last, most structural biologists and biochemists would interpret "protein instability" in a very different way and would think of decreased melting temperature, folding issues, or conformational heterogeneity, none of which was really tested in this study. Also, mutants could be affecting catalytic

activity allosterically, but this is not discussed. While these type of biochemical and biophysical experiments might be out of the scope of the present study, there should be clarity about the meaning of "instability" in this section and throughout the text.

As described in detail in 1), although used HEK293 cells instead, we compared the protein levels of wild-type PAF-AH2 and the mutants using the same pcDNA vector under the same conditions. These points that the reviewer pointed out have been corrected as follows.

Page 18, Line 14

“The forced expression of PAF-AH2 in BMMC was attempted with the plasmid vector but was unsuccessful. Therefore, HEK293 cells, which are human-derived cells that produced a sufficient amount of target protein via transfection of plasmid vector and had low endogenous production of PAF-AH2 protein, were selected as transfected cells.”

Page 26, Line 18

“PAF-AH2 expression vectors and transfection into HEK293 cells

We used the pcDNA3.1(+) plasmid vectors (V790202, Invitrogen) carrying a full length human *Pafah2* cDNA (NCBI reference sequence: NM_000437.4) cloned from the human brain cDNA library (Life Technologies, Inc.). Using the Quik Change Site-Directed Mutagenesis Kit (Agilent) according to the instruction manual, we introduced single-base substitution into the plasmid carrying native *Pafah2* cDNA to generate variants R85C (253C>T), Q184R (551A>G), and S236C (707C>G) and confirmed the presence of mutations by DNA sequencing. Native or mutant *Pafah2* pcDNA plasmids (3.5 µg per 6-well dish) were transfected into HEK293 cells using Lipofectamine 2000 (Invitrogen). After 48 hours, the culture medium was changed and exposed DMEM with or without MG132 (10µM) for 6 hours. Subsequently, cells were collected and the amount of expressed PAF-AH2 protein was analyzed by western blotting. HEK293 cells transfected by empty pcDNA vectors were used as controls.”

As the reviewer highlighted, we cannot rule out that the mutant protein allosterically affects the catalytic activity even if the mutation is not located in the active center. Also, as the meaning of instability is unclear, we have revised them as follows.

Page 18, Line 12

“Additionally, experiments using the expression vector in cultured cells revealed reduced expressed protein level associated with the two mutations. “

Page 18, Line 18

“Since treatment with a proteasome inhibitor restores the levels of the mutant proteins, we believe that post-translational modifications such as ubiquitination are involved in the reduction due to mutations.”

General comments:

- Page 2. No need to define the KO abbreviation in the abstract if it is not used elsewhere in the abstract.

In the revised manuscript, the KO abbreviation in the abstract was removed.

- Page 12. The sentence "Interestingly, treatment of MG132, ..." should probably be "Interestingly, treatment with MG132, ..."

We have revised it accordingly on page 12.

- Page 13. The sentence "Interestingly, the anti-fibrotic action and improvement of PH exerted by ω -3 epoxides observed in this study could not be confirmed when ω -3 fatty acids were administered at the same dose, so it is suggested that ω -3 epoxides possessed the structure-specific functions." is very unclear. Not sure what study the authors are referring to and what they are suggesting.

To assess whether the improvement in vascular remodeling is specific to ω -3 epoxides, we examined the ω -3 fatty acids (EPA corresponding to 17,18-EpETE, and DHA corresponding to 19,20-EpDPE) at the same concentration as the controls in Figures 4 and 5. The sentence was revised as follows.

Page 14, Line 15

"Interestingly, the anti-fibrotic action and improvement of PH exerted by ω -3 epoxides observed in this study could not be confirmed when ω -3 fatty acids (EPA and DHA) were administered at the same dose, suggesting that these functions were specific to ω -3 epoxides."

- Page 14. The sentence "..., it could be considered that mast cells released ω -3 epoxides in an extracellular manner." Isn't this redundant? (released to the extracellular space?)

We thank the reviewer for the suggestion. We have revised the text as follows.

Page 16, Line 11

"..., it could be considered that mast cells released ω -3 epoxides to the extracellular space."

Reviewer #4 (Remarks to the Author):

Reviewer Critique

In a study, entitled "Omega-3 fatty acid epoxides produced by PAF-AH2 in mast cells regulate pulmonary vascular remodeling", the authors addressed the role of specific medications that suppress pulmonary vascular remodeling in pulmonary hypertension (PH), searching for functional lipids involved in the pathophysiology of the condition. The authors found that epoxidized ω -3 fatty acids (ω -3 epoxides) and the expression of PAF-AH2, a key enzyme that releases ω -3 epoxides from membrane phospholipids in mast cells, were reduced in the lungs of hypoxic PH mouse model. Pafah2 knock-out (KO) mice exhibited the exacerbation of hypoxic PH via advanced pulmonary vascular remodeling, whereas ω -3 epoxides suppressed the activation of lung fibroblasts through controlling TGF- β signaling. In vivo supplementation with ω -3 epoxides reduced the progression of PH not only in hypoxic exposure, but also during hypoxia. Analysis of whole-exome sequencing data from patients with PAH uncovered two candidate pathogenic variants of Pafah2 resulting in protein degradation. The authors conclude that their findings support that PAF-AH2/ ω -3 epoxide production axis could be a promising therapeutic target for PH. While this study provides potential new insights into the role of PAF-AH2 and ω -3 epoxide in PH of potential interest to the readership of Nature Communications, this is a candidate gene study with modest signal that needs to be replicated and further validated to ensure biological and statistical significance of the biological pathway(s) involved in a larger study. I have outlined my concerns in the comments below:

Major Comments:

1. The authors show modest reduction Omega-3 epoxides in murine lungs with exposure of hypoxia at 4, 14 and 28 days in n=3 compared to normoxia; how were these results controlled for; was any alternative method to that of LC-MS/MS used to validate these results.

We thank the reviewer for the valuable suggestions.

Currently, LC-MS/MS is the best method for accurately measuring trace amounts of fatty acid metabolites. ELISA for measuring EpETE, EpDPE, or their dihydrodiols has not yet been developed.

For the lipidomics in the present study, using high-quality reference materials of the fatty acid metabolites, we determined the optimal MRM pair and parameters including retention time and formed a calibration curve to enable quantitative analysis. By using 11(12)-EET-d11 as an internal standard, the extraction efficiency and the variation between the measurement samples are fixed at the same time to improve the measurement accuracy.

We have added the details of measurement to the Methods section.

Page 20, Line 15

"Lipid extraction from the lung samples

Details of the lipid extraction were previously described. Murine right lung tissues were harvested and immediately placed in liquid nitrogen. After freeze crushing of samples, lipids were extracted by the method of Bligh and Dyer. Extracted solutions were dried up with centrifugal evaporator, dissolved in methanol : isopropanol=1:1, and stored at -20°C. Fatty acid metabolites were further purified from tissues by solid-phase extraction using InertSep NH₂ columns (GL Science) with deuterium-labelled internal standard (11(12)-EET-d11). Briefly, InertSep NH₂ columns were preconditioned with 6 ml of hexane and lipids extracted from tissues by the method of Bligh and Dyer were applied with 500µL of chloroform. Columns were then washed with 6 ml of chloroform/isopropanol (2/1, v/v), followed by the elution with diethyl ether/acetic acid (98/2, v/v). The extracted solutions were dried up with centrifugal evaporator, dissolved in methanol : isopropanol=1:1, and stored at -20°C.”

Page 21, Line 9

“ For the detection of fatty acid metabolites, LC/ESI-MS-based lipidomics analyses were performed on a Shimadzu Nexera UPLC system (Shimadzu) coupled with a QTRAP 4500 hybrid triple quadrupole linear ion trap mass spectrometer (AB SCIEX). Chromatographic separation was performed on a ACQUITY UPLC HSS T3 column (2.1×100 mm, 1.8 µm; Waters) maintained at 40°C using mobile phase A (water/acetic acid (100/0.1, v/v) containing 10 mM ammonium acetate) and mobile phase B (acetonitrile/methanol (4/1, v/v) containing 10 mM ammonium acetate) in a gradient program (0–2 min: 90% A; 2–10 min: 90% A →30% A; 10–24 min: 30% A →27% A; 24–27 min: 1% A; 27–32 min: 90% A) with a flow rate of 0.2 ml/min(0–10 min), 0.1 ml/min(10–15 min),0.2 ml/min(15–24 min) and 0.5 ml/min(24–32 min). The instrument parameters were as follows: curtain gas, 10 psi; ion spray voltage, -4500 V; temperature, 600°C; ion source gas 1, 70 psi; ion source gas 2, 80 psi. The specific detection was performed by MRM as described previously (*Gut.* 2021; 70(1): 180–193).”

2. The authors show that hypoxic pulmonary hypertension is exacerbated in *Pafah2* KO mice. Did the authors control for %KO/KD values by sequencing and mRNA expression and is there a control gene expression. As shown in figure 2, several of the effects are modest and in the absence of proper control gene profile may be hard to interpret.

Pafah2 KO mice were created by homologous recombination in embryonic stem cells (*J Biol Chem* 2008; 283: 1628-1636). The targeting vector was constructed to replace exons 8-9 of the *Pafah2* gene with a neomycin-resistance gene. Since homozygous *Pafah2* KO mice were used in the experiment, mRNA was not detected by qPCR using the primer designed for the deleted exon (see the figure on the right), and PAF-AH2 protein is not detected at all in WB (Figure 2a).

All the experimental results that evaluated the mRNA expression level in this study

were the $\Delta\Delta C_t$ value calculated by using ribosomal 18S as the house keeping gene. We have revised the captions for Figure 2g, 2i, 3h, 3i, 4c, and Supplementary Figure 6a, 7b, 8a, 9a, 9b.

3. What control staining did the authors use to ensure cells other than mast cells are not contributing to the observed modest changes in phenotypes observed in *Pafah2* KO mice and humans under hypoxic PH conditions; what about macrophages, fibroblasts, endothelial or epithelial cells.

We thank the reviewer for the valuable comment. Immunohistological staining for the markers of macrophages (mouse Mac3 or human CD68), fibroblasts (Vimentin), endothelial cells (CD31), and alveolar epithelial cells (Podoplanin) was performed on the lung tissue in hypoxic-PH mice and in the patients with PAH, but co-expression with PAF-AH2 was not observed in cells other than mast cells (Supplementary Figure 4a,b). Therefore, it was unlikely that these cells contributed to the phenotype of *Pafah2* KO mice in this study.

The manuscript has been revised as follows.

Page 7, Line 3:

“Fluorescent immunohistochemistry of lung samples from the hypoxic PH mouse model, Sugen/hypoxia PH mouse model, and human idiopathic PAH patients revealed that the cells expressing PAF-AH2 were tryptase-positive mast cells, but not macrophages, fibroblasts, endothelial cells, and respiratory epithelial cells (Figure 3a, Supplementary Figure 4a, 4b).”

4. The effects of omega-3 epoxides on suppression of cell activation and proliferation appears modest and vehicle alone follows same pattern for several of the measures; this should be presented in bar graph as % of control for better clarity.

We thank the reviewer for the valuable suggestions. For better clarity, the data in Figure 4 are now presented in bar graph as a percentage of the control.

5. The effects of ω -3 epoxides on improving hypoxic PH in mice/human similarly appear modest and unconvincing. This should be similarly presented as 5 control and ensure there is enough control comparison.

We re-conducted the experiment on the administration of ω -3 epoxides with a sufficient number of samples ($n>5$) in Supplemental Figure 10b-d, and thereafter confirmed that ω -3 epoxides showed a significant improvement for PH in both WT and *Pafah2* KO mice.

6. The potential role of *Pafah2* in the development of human PAH is very preliminary and unclear about the pathogenic significance of these two variants despite positive scores and how many patients carry them. Are these variants listed as pathogenic in ClinVar, HGMD or other databases, how many subjects carry these variants, how many subjects are in gnomAD or other large public databases who carry these variants and may not have PH.

We thank the reviewer for the important comment.

The two variants of PAF-AH2 in the present study were not listed in ClinVar and HGMD. Using SIFT and Polyphen, the prediction of the impact of missenses for the two mutations on the structure and function, resulted in “deleterious,” “probably damaging” outcome. In addition, the combined annotation dependent depletion (CADD PHRED) showed a significantly high score of 32 for both variants, indicating significant harm. The percentage of allele frequencies of these mutations in the global healthy population obtained from the gnomAD v2.1.1 (The Genome Aggregation Database) and the international large-scale gene database including a wide variety of population is 0.22 % (c.253C> T) and 0.0863 % (c.551A> G), respectively. The percentage of allele frequencies of each mutation in the Japanese general population obtained from ToMMo (Tohoku Medical Megabank Organization) 103.5KJSNV ver.1/2 data is also extremely low at 0.45 % (c.253C> T) and 0 % (c.551A> G). We summarized this information in Supplementary Table 2.

Minor comments:

1. In methods, should be separate section on human subjects describing their mutations and phenotypes and how these were validated.

We thank the reviewer for valuable suggestion.

We focused on the variants of *Pafah2* gene in the whole-exome sequencing data of patients with PH and selected missense mutations with a CADD score equal to or greater than 30 (MIS30). In the Methods section, we revised the description for whole-exome sequencing data of human patients with PH as follows.

Page 28, Line 14

“Whole-exome sequencing of human PH patients

This study has been approved by the Institutional Review Board (IRB) of author’s affiliation, and all genetic tests were performed with informed consent from patients after genetic counseling. We included and analyzed blood samples from 262 patients including 90 idiopathic PAH, 61 heritable PAH, and 54 connective tissue disease-associated PAH (mean age, 45.5±16.6 years; female, 78.0%; mean pulmonary artery pressure, 42.9±15.5 mmHg). We performed whole-exome sequencing using a HiSeq 2500 platform (Illumina, San Diego, CA) and SureSelectXT Human All Exon Kit (Agilent Technologies, Santa Clara, CA)

for hybridization capture. Pathogenicity of variants was assessed with CADD, SIFT, and PolyPhen-2. Regarding to variants of *Pafah2* gene in the whole-exome sequencing data of patients with PH, we selected missense mutations with a CADD score of equal to or greater than 30 (MIS30). Known pulmonary arterial hypertension (PAH)-related genes includes bone morphogenetic protein receptor type 2 gene (BMPR2), activin A receptor-like 1 gene (ACVRL1), endoglin gene (ENG), caveolin-1 gene (CAV-1), T-box 4 gene (TBX4), potassium channel subfamily K member 3 gene (KCNK3), eukaryotic initiation translation factor 2 a kinase 4 (EIF2AK4), SMADs, adenosine triphosphate 13A3 (ATP13A3), aquaporin 1 gene (AQP1), growth differentiation factor 2 gene (GDF2), and SRY-related high-mobility group box family member 17 gene (SOX17), which were selected according to the previous report.”

The in vitro assay performed to validate properties of mutant PAF-AH2 proteins was added to the Method section as follows.

Page 26, Line 18

“PAF-AH2 expression vectors and transfection into HEK293 cells

We used the pcDNA3.1(+) plasmid vectors (V790202, Invitrogen) carrying a full length human *Pafah2* cDNA (NCBI reference sequence: NM_000437.4) cloned from the human brain cDNA library (Life Technologies, Inc.). Using the Quik Change Site-Directed Mutagenesis Kit (Agilent) according to the instruction manual, we introduced single-base substitution into the plasmid carrying native *Pafah2* cDNA to generate variants R85C (253C>T), Q184R (551A>G), and S236C (707C>G) and confirmed the presence of mutations by DNA sequencing. Native or mutant *Pafah2* pcDNA plasmids (3.5 µg per 6-well dish) were transfected into HEK293 cells using Lipofectamine 2000 (Invitrogen). After 48 hours, the culture medium was changed and exposed DMEM with or without MG132 (10µM) for 6 hours. Subsequently, cells were collected and the amount of expressed PAF-AH2 protein was analyzed by western blotting. HEK293 cells transfected by empty pcDNA vectors were used as controls.”

REVIEWERS' COMMENTS

Reviewer #1 (Remarks to the Author):

Thanks for addressing concerns

Reviewer #2 (Remarks to the Author):

Thank you very much for this thorough revision of the manuscript. I have no further questions - and look forward to seeing this study published in Nature Communications.

Sincerely,

Karsten Weylandt

Reviewer #3 (Remarks to the Author):

The revised version of the manuscript titled "Omega-3 fatty acid epoxides produced by PAF-AH2 in mast cells regulate pulmonary vascular remodeling" addresses most of my critiques by clarifying methods for in vitro expression of wild-type and mutant versions of PAF-AH2 and by clarifying that homology models were used in simulations rather than experimentally obtained structures. The authors also improved the methods section involving molecular dynamics simulations.

Interestingly, the RMSD plots in Fig. S11 show that the mutants (RMSD < 3 for most of the trajectory) are more stable than the wild-type protein (RMSD > 3) during the simulated time (72 nano seconds), although the authors did little to explain or determine what was occurring at the molecular level. It is also unclear why the authors decided to show only a certain subset of frames in Fig. 6b, rather than others. This reviewer finds the analysis of simulations underwhelming, but satisfactory. A couple of errors need to be corrected and I do have some additional suggestions that the authors may want to consider to enhance their report:

1) Errors - The text states that simulations were run for "72 nano-seconds". If so, the caption of figure 6b stating "... (corresponding to 54-60 micro-seconds)" is erroneous. This should be 54-60 nano-seconds. Similarly, the caption of Supplementary Figure 11 states "... showing stabilization of the model structure in 60 micro seconds.". Please note that 60000 pico seconds is 60 nano seconds, not micro seconds.

2) Suggestions

- Citations to software and force-field used (NAMD, VMD, CHARMM) should be included along with a citation to the structure used for homology modeling (PDB code 5i9i is no enough, include citation).

- Can the authors clarify why they picked the time-window of 54-60 nano-seconds in Fig. 6b? Is this representative of what happens in the remainder of the trajectory?

- What was used as a reference to compute RMSD in each trajectory?

- If the same reference (original model?) was used for RMSD calculations, it seems interesting that the mutant proteins are more stable than the wild type model (RMSD below 3Å for mutants). This partially supports the claim that mutants are behaving differently than the wild type. Can the authors inspect the trajectory to explain why? Whether this is meaningful in a short 72 nano-second trajectory is debatable, but could be interesting. How is this related to ubiquitination is very unclear.

- Page 18 states "Since treatment with a proteasome inhibitor restores the levels of mutant proteins," should be "Since treatment with a proteasome inhibitor partially restores the levels of mutant proteins," to be consistent with the data and the statement in page 12, line 18.

Reviewer #4 (Remarks to the Author):

The authors have been responsive to the comments raised which has resulted in a significantly improved manuscript. I have no further comments.

Reviewer #3 (Remarks to the Author):

The revised version of the manuscript titled "Omega-3 fatty acid epoxides produced by PAF-AH2 in mast cells regulate pulmonary vascular remodeling" addresses most of my critiques by clarifying methods for in vitro expression of wild-type and mutant versions of PAF-AH2 and by clarifying that homology models were used in simulations rather than experimentally obtained structures. The authors also improved the methods section involving molecular dynamics simulations.

Interestingly, the RMSD plots in Fig. S11 show that the mutants (RMSD < 3 for most of the trajectory) are more stable than the wild-type protein (RMSD > 3) during the simulated time (72 nano seconds), although the authors did little to explain or determine what was occurring at the molecular level. It is also unclear why the authors decided to show only a certain subset of frames in Fig. 6b, rather than others. This reviewer finds the analysis of simulations underwhelming, but satisfactory. A couple of errors need to be corrected and I do have some additional suggestions that the authors may want to consider to enhance their report:

We thank the reviewer for the constructive comments. In the revised manuscript, we have modified the manuscript that should satisfactorily address the concerns of the reviewers.

1) Errors - The text states that simulations were run for "72 nano-seconds". If so, the caption of figure 6b stating "... (corresponding to 54-60 micro-seconds)" is erroneous. This should be 54-60 nano-seconds. Similarly, the caption of Supplementary Figure 11 states "... showing stabilization of the model structure in 60 micro seconds.". Please note that 60000 pico seconds is 60 nano seconds, not micro seconds.

We thank the reviewer for the important point and we apologize for the misrepresentation. We have changed the sentences in the figure legend as below:

Figure 6 (b)

..., showing the stacked snapshots of 300 frames obtained from frame 2700–3000 (corresponding to 54–60 nano-seconds).

Supplementary Figure 11

Root mean square deviation (RMSD) plot showing stabilization of the model structure in 60 nano-seconds.

2) Suggestions

- Citations to software and force-field used (NAMD, VMD, CHARMM) should be included along with a citation to the structure used for homology modeling (PDB code 5i9i is no enough, include citation).

We thank the reviewer for the important point. We have added the citations for NAMD, VMD, CHARMM, and PDB, according to your suggestions.

- Can the authors clarify why they picked the time-window of 54-60 nano-seconds in Fig. 6b? Is this representative of what happens in the remainder of the trajectory?

The time-window of 54-60 nano-seconds was chosen as a time-window representing structural fluctuations after the model stabilized (as evident in the stabilization of RMSD shown in Supplementary Figure 11). Without external force, we think that the time-points after model stabilization will be within the range of thermal fluctuation and thus think that this time-window is representative of the remainder of the trajectory.

We have added to the point in the method section:

Page 34, line 10

The stacked snapshots of 300 frames obtained from frame 2700–3000 (corresponding to 54–60 nano-seconds), a time-window representing structural fluctuations after the model stabilized, were shown as figures.

- What was used as a reference to compute RMSD in each trajectory?
- If the same reference (original model?) was used for RMSD calculations, it seems interesting that the mutant proteins are more stable than the wild type

model (RMSD below 3Å for mutants). This partially supports the claim that mutants are behaving differently than the wild type. Can the authors inspect the trajectory to explain why? Whether this is meaningful in a short 72 nano-second trajectory is debatable, but could be interesting. How is this related to ubiquitination is very unclear.

The RMSD was calculated with distances relative to the initial model. While the initial model of the wild-type was the immediate output of the homology modeling solvated in water, the initial models for the mutants were the stable structure of the wild-type (after 60 nano-second simulation).

We thank the reviewer for the good suggestions, but as mentioned above, each initial model used as a reference for calculating RMSD is different, so it is not possible to directly compare the RMSD between the wild-type and the mutants. To avoid misunderstanding, we have added the contents regarding the initial models in the method section as follows:

Page 34, line 3

The RMSD was calculated with distances relative to the initial model. While the initial model of the wild-type was the immediate output of the homology modeling solvated in water, the initial models for the mutants were the stable structure of the wild-type (after 60 nano-second simulation).

- Page 18 states "Since treatment with a proteasome inhibitor restores the levels of mutant proteins," should be "Since treatment with a proteasome inhibitor partially restores the levels of mutant proteins," to be consistent with the data and the statement in page 12, line 18.

We thank the reviewer for the important comments. We have modified the sentences according to your suggestions (page 12, line 18).